



**An improved dynamic bidirectional coupled hydrologic-**
**hydrodynamic model for efficient flood inundation prediction**
Yanxia Shen, Zhenduo Zhu, Qi Zhou, Chunbo Jiang[*]
State Key Laboratory of Hydroscience and Engineering, Department of Hydraulic
Engineering, Tsinghua University, Beijing, 100084, China
**Abstract:** To improve computational efficiency while maintaining numerical accuracy,
coupled hydrologic-hydrodynamic models based on non-uniform grids are used for
flood inundation prediction. In those models, a hydrodynamic model using a fine grid
can be applied for flood-prone areas, and a hydrologic model using a coarse grid can
be used for the rest of the areas. However, it is challenging to deal with the separation
and interface between the two types of areas because the boundaries of the flood-prone
areas are time-dependent. We present an improved Multigrid Dynamical Bidirectional
Coupled hydrologic-hydrodynamic Model (IM-DBCM) with two major improvements:
1) automated non-uniform mesh generation based on the D∞ algorithm was
implemented to identify the flood-prone areas where high-resolution inundation
conditions are needed; 2) ghost cells and bilinear interpolation were implemented to
improve numerical accuracy in interpolating variables between the coarse and fine grids.
A hydrologic model, two-dimensional (2D) nonlinear reservoir (NLR) model was
bidirectionally coupled with a 2D hydrodynamic model that solves the shallow water
equations. Three cases were considered to demonstrate the effectiveness of the
improvements. In all cases, the mesh generation algorithm was shown to efficiently and
successfully generate high-resolution grids only in those flood-prone areas. Compared
with the original M-DBCM (OM-DBCM), the new model had lower RMSEs and higher

[*]Corresponding author: State Key Laboratory of Hydroscience and Engineering, Department of Hydraulic Engineering, Tsinghua University, Beijing, 100084, China
Corresponding author: Tel: +8613581891886; E-mail address: jcb@tsinghua.edu.cn



NSEs, indicating that the proposed mesh generation and interpolation were reliable and
stable. It can be adapted adequately to the real-life real flood evolution process in
watersheds and provide practical and reliable solutions for rapid flood prediction.
**Key words:** Coupled hydrologic-hydrodynamic model; Multi-grid generation; Bilinear
interpolation; Computational efficiency and accuracy; Flood simulation
**1 Introduction**
Floods are the most frequent natural disasters that seriously harm human health
and economic growth. Numerical models are critical for predicting flooding processes
to help prevent or mitigate the damaging effects of floods on people and communities
(Bates, 2022). Coupled hydrologic-hydrodynamic models are widely used to translate
the amount of rainfall obtained from weather forecasting models or rain gauge
observations into surface inundation (Xia et al., 2019).
Coupled hydrologic-hydrodynamic models can be generally divided into two
categories. The first category is full 2D hydrodynamic models (HM2D), where the 2D
hydrodynamic model is used to simulate the overland flow (runoff routing and flood
inundation), and only the runoff generation is calculated by the hydrologic model and
added into the mass source term of the 2D hydrodynamic model (Singh et al., 2011;
Garcia-Navarro et al., 2019; Hou et al., 2020; Costabile and Costanzo, 2021). The
development and simulation of HM2D require high-resolution topographic data at the
catchment scale and extensive computational time, which hinder their application in
large-scale flood forecasting. A promising way to achieve computational speedups is
execution in a massively parallel fashion on supercomputers (Noh et al., 2018; Kuffour
et al., 2020) or graphic processing unit (GPU) (Kuffour et al., 2020; Ming et al., 2020;
Morales-Hernández et al., 2021; Buttinger-Kreuzhuber et al., 2022). Besides the
computational efficiency, the numerical stability of HM2D can be problematic,





especially in thin-layer water regions (Kim et al., 2012).

The second category of coupled hydrologic-hydrodynamic models uses

hydrologic models for upland areas and hydrodynamic models for main channels and
flood-prone areas, and the hydrologic model provides the discharge boundary for the
hydrodynamic model (Hdeib et al., 2018; Munar et al., 2018; Shin et al., 2019; Gomes,
et al., 2021). And therefore, this category can be more efficient and thus applied to
large-scale studies. This category is further divided into one-way and two-way coupling
models, according to whether the hydrodynamic model provides feedback to the
hydrologic model.

In one-way coupling models, the hydrologic model is run first and independently

from the hydrodynamic model. The hydrographs obtained from hydrologic models are
used as an input for the hydrodynamic models in a fixed position (Schumann et al.,
2013; Feistl et al., 2014; Choi and Mantilla, 2015; Bhola, 2018; Wing et al., 2019). This
one-way flow information cannot capture the mutual interaction between runoff
production and flood inundation, and the fixed interface is inconsistent with the actual
flood process where the inflow discharge positions, flow path, and discharge values
change with accumulating rainfall.

In two-way coupling models, the governing equations of the hydrologic and

hydrodynamic models are solved simultaneously in each time step, with information at
the shared interface updated and exchanged at each or several computational time steps.
Most existing two-way models are the coupling of hydrologic and one-dimensional (1D)
hydrodynamic models, such as the coupling of Mike SHE and Mike 11, SWMM
(Thompson et al., 2004; Laganier et al., 2013; Rossman, 2015; Chalkidis et al., 2016).
The application of 1D modeling of overland flow is limited when developing precise
and reliable flood maps in 2D inundation regions. Jiang et al. (2021) proposed a




dynamic bidirectional coupling model (DBCM), where the hydrologic and 2D
hydrodynamic models were solved synchronously in each time step. The hydrologic
and 2D hydrodynamic models are coupled through a coupling moving interface (CMI),
and the inflow discharge positions and flow path change with accumulating rainfall, it
had better numerical accuracy than the one-way coupling models.
However, because uniform grids were adopted in DBCM, high-resolution
simulations in a large domain inevitably involved numerous computational nodes and
substantially increased the computational cost. An essential consideration to reduce
computational time is mesh coarsening (Caviedes-Voullième et al., 2012). Adaptive
mesh refinement (AMR) has been used to optimize the grid resolution during flood
simulations (Donat et al., 2014; Hu et al., 2018; Ghazizadeh, 2020; Ding et al., 2021;
Kesserwani and Sharifian, 2023). Aiming to increase computational efficiency by
reducing computing nodes, it adjusts grid size for local grid refinement by domain
features or flow conditions. Yu (2019) used quadtree grids to divide the computational
domain and applied the DBCM to simulate the flooding process. It needs to segment
and merge the grid elements repeatedly during the calculation, which can be time-
consuming and offset the calculation time saved by the optimized grid. Besides, the
mesh generation and flood simulation were compiled in the same code base, which
increased the computational cost and total execution time.
Static non-uniform grids have increasingly received attention in recent years,
which simplified grid generation procedure compared with AMR (Caviedes-Voullième
et al., 2012; Hou et al., 2018; Bomers et al., 2019; Ozgen-Xian et al., 2020). Compared
with uniform grids and AMR, it can not only reduce computational nodes, but use
different time steps in different grid sizes to further reduce computation time. Shen et
al. (2021) and Shen and Jiang (2023) divided the computational domain based on static



multi-grids, where the different grid size ratios of coarse to fine grids were designed.
But there were two limitations to this scheme. One limitation is that the grids need to
be generated manually, which can be subjective and uncertain. It also needs a heavy
workload, especially for large watersheds. Besides the grid generation, the variable
interpolation between the coarse and fine cells was also not reasonable. There are
shared and hanging nodes at the interpolation interface. Shen et al. (2021) assumed the
variables at the shared nodes were equal to that at the cell center, and the hanging nodes
were obtained from shared nodes. The results showed that this scheme has
unsatisfactory accuracy and frequently fails to converge. Although the multi-grid-based
model can reduce computational time, there are remarkable challenges such as the grid
partition technique, determination of coarse and fine regions, and variables
interpolation between coarse and fine grids.

The objective of this study is to develop an integrated system that fully couples

the hydrologic and 2D hydrodynamic models, utilize an automated method for efficient
multi-grid mesh generation, and resolve variable interpolation between coarse and fine
grids more accurately. An improved dynamic bidirectional coupling model (IM-DBCM)
was presented, where the 2D nonlinear reservoir (NLR) model was coupled with the
full 2D hydrodynamic model through a CMI. The D∞ algorithm was implemented to
divide the computational domain into non-uniform grids automatically. Ghost cells and
bilinear interpolation were used to interpolate variables between the coarse and fine
grids. Three case studies, two laboratory experiments and one real-world watershed,
were conducted, and the simulation results were compared with the original M-DBCM
(OM-DBCM) to evaluate the effectiveness of the improvements.
**2 Methodology**

The Fortran programming language was adopted to apply the coupling model. The





framework of IM-DBCM is illustrated in Figure 1. The model consists of two
components: a hydrologic model (i.e., 2D NLR model) that simulates the runoff
generation and routing, and 2D hydrodynamic model simulating the flood inundation
process. Before the model setup, it is required to first design the grids. For the model
execution, the variables interpolation between coarse and fine grids and the coupling of
hydrologic and hydrodynamic models are the two main issues that must be addressed.

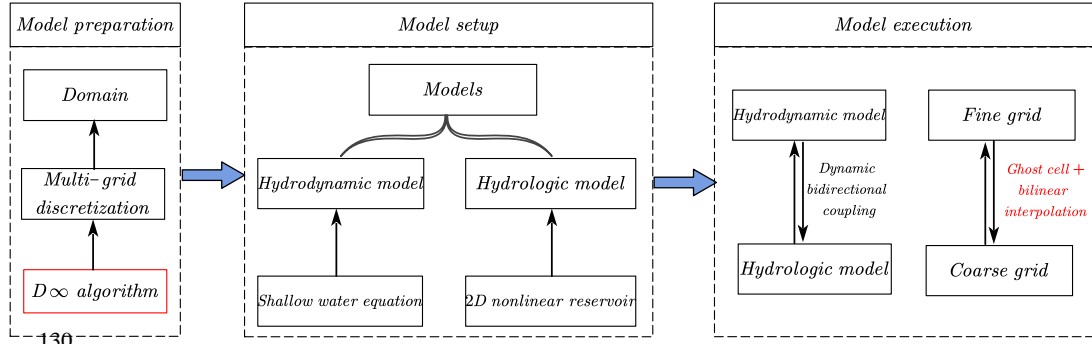


Figure 1 Framework of IM-DBCM

**2.1 Automated multi-grid generation**

Associated with flood models, the design of computational grids that are scalable

and suitable for all applications is challenging. The grid generation can be considered
as a model preprocess, which is the foundation of flood simulation and can influence
both computational accuracy and efficiency. In this study, a multi-grid generation
method was proposed based on the D∞ algorithm, to generate refined grid cells at flood-
prone areas where high-resolution representation of topographic features is essential for
flood simulation while discretizing the rest of the domain using coarse grids. The D∞
algorithm is a method of representing flow directions based on triangular facets in grid
DEM proposed by Tarboton (1997). It allocates the flow fractionally to each lower
neighboring grid in proportion to the slope toward that grid. The flow direction is
determined as the direction of the steepest downward slope on the eight triangular facets



formed across a $3 \times 3$-pixel window centered on the pixel of interest, which was detailed
by Tarboton (1997). Compared with the D8 algorithm, where the flow is discretized
into only one of eight possible directions, separated by 45°, the D∞ algorithm is more
reasonable and accurate for delineating the actual river trend.

The process of discretizing computational domain based on the D∞ algorithm is

shown in Figure 2. First, a raw DEM was prepared, and sink filling was performed on
the DEM. Second, the D∞ algorithm was applied to determine the flow direction on
grids. Subsequently, the upslope area, defined as the total catchment area that is
upstream of a grid center or short length of contour (Moore et al., 1991), was calculated
based on the flow direction. Finally, an area threshold was defined to identify the slope
lands and derive the river drainage networks from accumulated drainage areas. In a grid
cell, if the upslope area was larger than the predefined threshold, it was considered as a
river drainage network; otherwise, it was defined as slope lands. The generated slope
lands and river network were verified through field surveys or satellite images-based
estimates. Generally, the river drainage networks present low slopes and hydraulic
conveyance, which is subject to flooding. Therefore, these areas should be discretized
using fine grids to represent the flooding process in high resolution. However, in the
slope lands, fine grids were not required and coarse grids were used to improve
computational efficiency. Because the regions of interest were of high resolution, the
reliability of the prediction would not deteriorate, although the number of grid cells was
considerably reduced, which can increase model efficiency and capability for flood
simulations over large domains. Compared with manual work, the grid generation
based on the D∞ algorithm can both reduce workload and time.



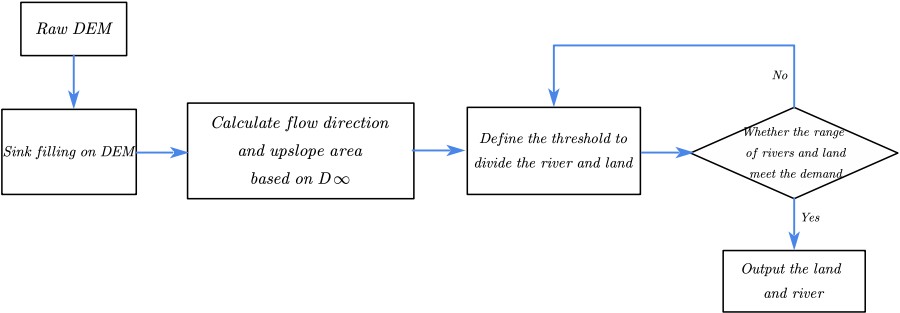


Figure 2 Grid generation based on the D∞ algorithm

A schematic of grid generation is shown in Figure 3. Two types of connecting

interfaces are presented, which divide the computing domain into three parts. The first

type is the red line (Variable Interpolation Interface, VII) between the coarse and fine

grids. The grid cell size changes suddenly on both sides of this line. The second type

(Coupling Moving Interface, CMI) is marked in blue on fine grids, which is moving

and time-dependent. The first part represents the coarse-grid areas, where the

hydrologic model is used to simulate rainfall-runoff. The other two parts are located in

the fine-grid areas. The regions between VII and CMI are defined as intermediate

transition zones, where the hydrologic model is used to simulate the flooding process.

These transition zones facilitate the application of different time steps in different grid

cell sizes  to improve computational efficiency. The hydrologic and hydrodynamic

models are dynamically coupled to represent the flooding process on fine grids, and the

CMI is a coupling boundary.





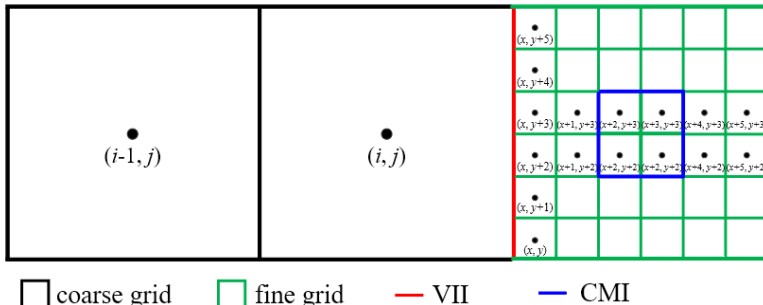


Figure 3. Schematic diagram of grid generation, where *i* and *j* are the coordinates of
coarse grid; *x* and *y* are the coordinates of fine grid; VII is the Variable Interpolation

Interface and CMI is the Coupling Moving Interface

**2.2 Variable interpolation between coarse and fine grids**

During a flow computation, if a cell has a neighbor of different size, interpolation

may be required to approximate variables in certain locations so that the governing
equation can be solved smoothly. An example is presented in Figure 4(a), where the
coarse grid has two eastern neighbors that are half its size. In this case, the variable
values of the smaller cells are obtained from those of larger cells. In the traditional
method, these variables are directly calculated using certain interpolation methods.
There are shared ($P_1$, $P_2$) and hanging ($Q$) nodes at the interface between the coarse and
fine grids. In Shen et al. (2021), the variable values on shared nodes can be transmitted
directly, while the values on hanging nodes were obtained by linear interpolation of the
shared nodes. This method is simple, feasible and easy to use. However, the variable
values are stored at the cell center, and there are no values at the interface nodes. Shen
et al. (2021) assumed that the values at the interface nodes were equal to that at the cell
center. It is inaccurate to make such an assumption, which can bring errors. And the
resulting error will increase as the cell size increases.

To overcome these drawbacks, ghost cells and bilinear interpolation method were



used to interpolate variables between coarse and fine grids. Figure 4(a) shows the
variable interpolation between the coarse and fine grids. Two ghost fine cells were
created, which were overlaid with partial coarse grids. The variables on the ghost fine
cells were interpolated through the coarse and fine grids between the interface, which
were then used as the boundary conditions for the calculation of the fine grids at the
next time step. The bilinear interpolation method was applied. The variable
interpolation may involve variables at locations $c_1$ , $c_2$ , $c_3$ , $f_{v1}^{'}$, $f_{v2}^{'}$, $f_1$ and $f_2$ . As the
variables are stored at the cell center, the variables at $c_1$ , $c_2$ , $c_3$ , $f_1$ and $f_2$ are available
directly. The values at $f_{v1}^{'}$ and $f_{v2}^{'}$ are obtained via natural neighbor interpolation, as
follows:

$$U_{f_{v1}^{'}} = U_{c_1} + \frac{U_{c_2} - U_{c_1}}{y_{c_2} - y_{c_1}}(y_{f_{v1}^{'}} - y_{c_1}) \tag{1}$$

$$U_{f_{v2}^{'}} = U_{c_3} + \frac{U_{c_1} - U_{c_3}}{y_{c_1} - y_{c3}}(y_{f_{v2}^{'}} - y_{c_3}) \tag{2}$$

where $U_{f_{v1}^{'}}, U_{f_{v2}^{'}}, U_{c_1}, U_{c_2}, U_{c_3}$ are the variables at locations $f_{v1}^{'}, f_{v2}^{'}, c_1, c_2, c_3$ respectively;
$y_{f_{v1}^{'}}, y_{f_{v2}^{'}}, y_{c_1}, y_{c_2}, y_{c_3}$ are the coordinates in $y$ directions at $f_{v1}^{'}, f_{v2}^{'}, c_1, c_2, c_3$ respectively.
And then, the variables of ghost fine cells at $f_{v1}$ and $f_{v2}$ can be calculated based
on that at $f_{v1}^{'}$ and $f_{v2}^{'}$, as follows:

$$U_{f_{v1}} = U_{f_{v1}^{'}} + \frac{U_{f_1} - U_{f_{v1}^{'}}}{x_{f_1} - x_{f_{v1}^{'}}}(x_{f_{v1}} - x_{f_{v1}^{'}}) \tag{3}$$

$$U_{f_{v2}} = U_{f_{v2}^{'}} + \frac{U_{f_2} - U_{f_{v2}^{'}}}{x_{f_2} - x_{f_{v2}^{'}}}(x_{f_{v2}} - x_{f_{v2}^{'}}) \tag{4}$$

where $U_{f_{v1}}, U_{f_{v2}}$ are the variables of ghost fine cells; $U_{f1}, U_{f_2}$ are the variables at $f_1$ , $f_2$ ,
respectively, which were calculated in the last time step; $x_{f_1}, x_{f_2}, x_{f_{v1}}, x_{f_{v2}}, x_{f_{v1}^{'}}$ and $x_{f_{v2}^{'}}$





are the coordinates in $x$ directions at $f_1$ , $f_2, f_{v1}, f_{v2}, f_{v1}', f_{v2}'$ respectively.

The values at $f_{v1}$, $f_{v2}$ were used as the boundary conditions for the calculation of

fine grids.

The variable interpolation from fine to coarse grids is presented in Figure 4(b),

where one ghost coarse cell was established. The variables of ghost coarse cells were
determined according to the fine and coarse grids between the interface. The variable
interpolation may involve variables at locations $c_v', c_1, f_1, f_2$ . As the variables are stored
at the cell center, the variables at $c_1, f_1, f_2$ are available directly. The values at $c_v'$ are
obtained via natural neighbor interpolation, as follows:

$$U_{c_v'} = U_{f_2} + \frac{U_{f_1} - U_{f_2}}{y_{f_1} - y_{f_2}}(y_{c_v'} - y_{f_2}) \tag{5}$$

where $U_{c_v'}, U_{f_1}, U_{f_2}$ are the variables at $c_v', f_1, f_2$ respectively; $y_{c_v'}, y_{f_1}, y_{f_2}$ are the
coordinates in $y$ direction at $c_v', f_1, f_2$ respectively.

And then, the variables of ghost coarse cells at $c_v$ can be calculated based on that

at $c_v', c_1$ , as follows:

$$U_{c_v} = U_{c_v'} + \frac{U_{c_1} - U_{c_v'}}{x_{c_1} - x_{c_v'}}(x_{c_v} - x_{c_v'}) \tag{6}$$

where $U_{c_v}$ are the variables of ghost fine cells; $U_{c_1}$ are the variables at $c_1$ ,which were
calculated in the last time step; $x_{c_1}, x_{c_v'}, x_{c_v}$ are the coordinates in $x$ direction at $c_1, c_v', c_v$
respectively.

The values at $c_v$ were used as boundary conditions for the calculation of coarse

grids at the next time step.





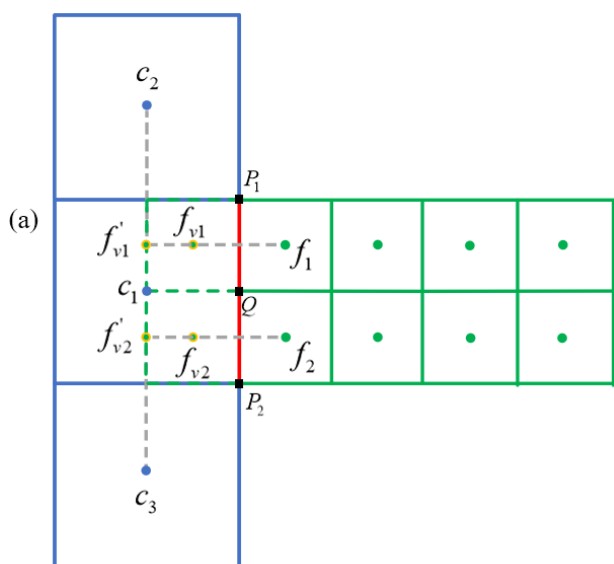


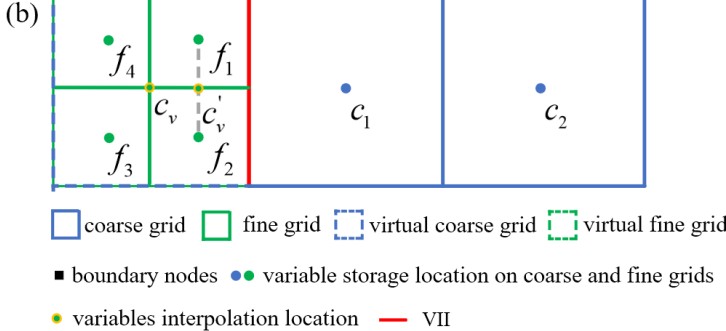


Figure 4. Variables interpolation between coarse and fine grids: (a) from coarse to

fine grids and (b) from fine to coarse grids

## 2.3 Numerical models

### 2.3.1 Hydrologic model

In this study, referring to the runoff calculation in the Storm Water Management

Model (SWMM), a 2D NLR model, including water balance and Manning equations,
was used to simulate rainfall-runoff. In SWMM, the watershed is divided into many
water tanks or reservoirs, where 1D NLR model including water balance and 1D
Manning equations is used to simulate the runoff (Rossman, 2015). It is a simple and



efficient method to calculate the runoff routing. In reality, however, the runoff routing
is a 2D way, so it is not accurate to calculate the 2D runoff routing using 1D NLR model.
Also, it is difficult to directly couple the 1D NLR model with 2D hydrodynamic model.
Therefore, the 2D NLR model was used to simulate the 2D surface runoff routing in
this study, as shown in Eqs. (7-11).

$$\frac{V_i^{n+1} - V_i^n}{\Delta t} = \left(Q_x\right)_{in\ i} - \left(Q_x\right)_{out\ i} + \left(Q_y\right)_{in\ i} - \left(Q_y\right)_{out\ i} + A_i q_{r\ i}^n \tag{7}$$

$$\left(Q_x\right)_{in\ i} - \left(Q_x\right)_{out\ i} = -\sum_{l=1}^{L} \left(q_{x\ \Gamma}^n \cdot n_x\right)_l \Delta L_l \tag{8}$$

$$\left(Q_y\right)_{in\ i} - \left(Q_y\right)_{out\ i} = -\sum_{l=1}^{L} \left(q_{y\ \Gamma}^n \cdot n_y\right)_l \Delta L_l \tag{9}$$

$$q_x = \frac{h^{5/3} S_x^{1/2}}{n_r} \tag{10}$$

$$q_y = \frac{h^{5/3} S_y^{1/2}}{n_r} \tag{11}$$

where the superscript $n$ and $n+1$ is the time step; $V$ is the water volume of grid (m$^3$);
$\left(Q_x\right)_{in\ i}, \left(Q_x\right)_{out\ i}$ is the inflow and outflow of grid $i$ in $x$ direction (m$^3$/s);
$\left(Q_y\right)_{in\ i}, \left(Q_y\right)_{out\ i}$ is the inflow and outflow of grid $i$ in $y$ direction (m$^3$/s); $q_{r\ i}$ indicates
runoff rate of grid $i$ (mm/h), which is rainfall intensity minus infiltration rate; $A_i$ is the
area of grid $i$ (m$^2$); $q_x, q_y$ are the unit discharge stored at cell-center along $x$ and $y$
direction (m$^2$/s), with $h$, $u$ and $v$ being water depth (m), flow velocity (m/s) in $x$ and $y$
directions, respectively; $q_{x\ \Gamma}, q_{y\ \Gamma}$ are the unit discharge at grid boundary in $x$ and $y$
direction, respectively (m$^2$/s), which are calculated based on $q_x, q_y$; $\Delta L_l$ is the side
length of grid (m); $l = 1, 2, 3, \dots, L$ is the number of edges of cell; $n_r$ is the Manning
roughness coefficient; $S_x$ and $S_y$ are water level gradients along $x$ and $y$ direction,



respectively, $S_x = -\dfrac{\partial}{\partial x}(z_b + h), S_y = -\dfrac{\partial}{\partial y}(z_b + h)$, where $z_b$ is the surface elevation.
**2.3.2 Hydrodynamic model**
The 2D shallow water equations (SWEs), consisting of mass and momentum
conservation equations (Toro 2001), were used to represent the hydrodynamic model.
The Godunov-type fine volume scheme with a Harten-Lax-van Leer contact (HLLC)
approximate Riemann solver (Toro et al., 1994) was adopted to solve the 2D SWEs,
and the second-order accuracy in temporal and spatial discretization was obtained based
on the Runge-Kutta method and Monotone Upstream-centered Schemes for
Conservation Laws (MUSCL) (Van Leer, 1979). The solution of SWEs was detailed in
many references (Toro 2001; Jiang et al., 2021).
**2.4 Dynamic bidirectional coupling of hydrologic and hydrodynamic models**
The hydrologic and hydrodynamic models were coupled dynamically and bi-
directionally. A water depth threshold was defined in advance and used to determine
the state of the cell. In a grid cell, if the water depth was lower than the predefined
threshold, it was defined as a non-inundation region where the hydrologic model was
applied. Conversely, if the water depth was higher than the threshold, it was considered
an inundation region where the 2D hydrodynamic model was applied. When the rainfall
intensity increased, the water depth increased because of the gradual accumulation of
surface water volume. Once the water depth exceeds the predefined threshold, the non-
inundation regions defined last time step may change to the inundation regions. The
inflow discharge positions, flow path, and discharge values subsequently changed, as
shown in Figure 5. Therefore, a CMI was formed between the inundation and non-
inundation regions. The hydrologic and 2D hydrodynamic models were coupled bi-
directionally, and the coupling interface was moving and time-dependent. The key issue
with the coupled model was to establish a reasonable approach for determining the



fluxes passing through the coupling interface, which should integrate the effect of the
current flow state obtained from these two models on both sides of the coupling
interface. The coupling method was described by Jiang et al. (2021).

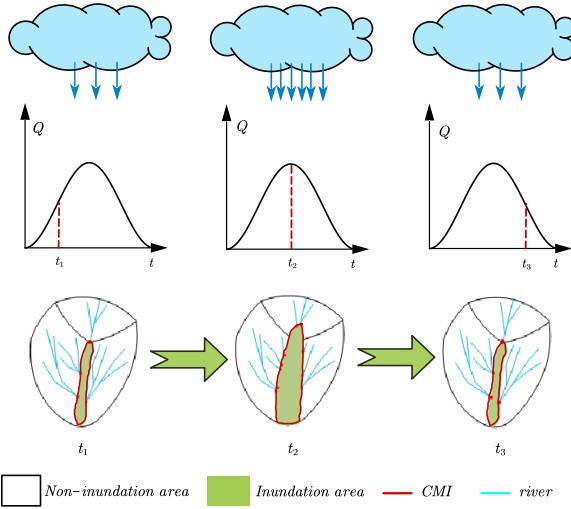

Figure 5. Variation in non-inundation and inundation regions with changing rainfall

conditions

**2.5 Time step**

An explicit scheme was used to solve the hydrologic and hydrodynamic models

over time. The time step was constrained by the Courant-Friedrichs-Lewy condition
(Delis and Nikolos, 2013), where the time step was a dynamic adjustment based on the
velocity and water depth in the computational domain. Different time steps were
adopted for the coarse and fine grids, and the time step of the fine grids was determined
as follows:
$$\Delta t_f = C \cdot min\left( \frac{min\left(\Delta x_f\right)}{max\left(|u_f| + \sqrt{gh_f}\right)}, \frac{min\left(\Delta y_f\right)}{max\left(|v_f| + \sqrt{gh_f}\right)} \right)$$
(12)

where $\Delta t_f$ is the time step of fine grids; $C$ is a constant used to maintain format stability;





$\Delta x_f$ and $\Delta y_f$ are the side lengths of fine grid in $x$ and $y$ directions; $u_f$ and $v_f$ are the
flow velocities on fine grids along $x$ and $y$ directions, respectively; $h_f$ is the water depth
on fine grids.
The time step of the coarse grids ($\Delta t_c$) was determined based on that of the fine
grids. If the size of the coarse grid was $k$ times that of the fine grid, the time step of the
coarse grid was determined to be $\Delta t_c = k\Delta t_f$.
**3 Results**
The performance of the IM-DBCM was analyzed by applying it to two 2D rainfall-
runoff experiments and one real-world flooding process. And the OM-DBCM
developed by Shen et al. (2021) was applied to the same cases for comparison with the
IM-DBCM.
**3.1 Rainfall over a plane with varying slope and roughness**
In this case, a sloping plan measuring $500 m \times 400 m$ was designed, with slopes
$S_{ox} = 0.02 + 0.0000149x$ and $S_{oy} = 0.05 + 0.0000116y$ along the $x$ and $y$ directions,
respectively (Jaber and Mohtar, 2003). The Manning coefficient is equal to
$n = \sqrt{n_x^2 + n_y^2}$, where $n_x = 0.1 - 0.0000168x$ and $n_y = 0.1 - 0.0000168y$. The rainfall
intensity is given by a symmetric triangular hyetograph $r = r(t)$, with
$r(0) = r(200\min) = 0$ and $r(100\min) = 0.8 \times 10^{-5} m/s$. The total simulation time was
14,400 s.
Different cases with various grid resolutions were developed to divide the
computational domain based on the D∞ algorithm, as listed in Table 1. In these cases,
the size of all the fine grids was $1m \times 1m$. The grid discretization of different cases is
shown in Figure S1 in Supplement.

Table 1 Different cases designed to simulate

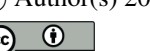



| Cases | The ratio of coarse to fine grids | Number of grids |
|-------|-----------------------------------|-----------------|
| case12 | 1:2 | 112,100 |
| case15 | 1:5 | 86,840 |
| case10 | 1:10 | 83,220 |

337  The hydrographs at the outlet node of coordinates of (500m, 400m) obtained from

338 different models were shown in Figure 6. A model proposed by Jaber and Mohtar (2003)

339 was also used to simulate the overland runoff. Because very fine grids and small time

340 step were used to divide the computational domain to obtain more accurate results in

341 the model developed by Jaber and Mohtar (2003), the results calculated by Jaber and

342 Mohtar (2003) can be used as a reference solution.

343  From Figure 6, the IM-DBCM held a shape close to the results simulated by Jaber

344 and Mohtar (2003) in all cases, as well as the peak discharge. But the peak discharge

345 of the hydrograph is slightly overestimated by the OM-DBCM, which may be attributed

346 to the difference in the variable interpolation between the coarse and fine grids. In the

347 OM-DBCM, variables at the interpolation interface were equal to that at the cell center,

348 which was then used to interpolate variables between the coarse and fine grids through

349 shared and hanging nodes. This interpolation method had two drawbacks. Firstly, it is

350 not reasonable to assume the variables at the interpolation interface are equal to that at

351 the cell center, and the resulting error could increase as the grid size increases. Besides,

352 compared with bilinear interpolation, the values at the hanging nodes are calculated by

353 linear interpolation through shared nodes, which may result in relatively large errors.

354 The results show that the methods to interpolate variable between the coarse and fine

355 grids by developing ghost cells proposed in this study has acceptable accuracy.

356  To quantitatively assess the performance of IM-DBCM, the Root Mean Square

357 Error (RMSE) of different cases was computed. The RMSEs of case12, case15 and

358 case10 were 4.01E-04, 7.85E-03 and 3.25E-02, respectively. It is shown that the error

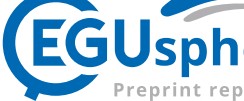



gradually increased with the increasing of the ratio of coarse to fine grids. The IM-
DBCM may capture the shape of the hydrograph in case12 and case15, both in limbs
and peak discharge, but the peak discharge is slightly underestimated in case10. A
possible explanation is that, compared to the coarse grids, the fine grids could better
capture the geometry of the channel cross-sections. High-resolution grids can better
represent small-scale topographic features and flow passages (Hou et al., 2018);
consequently, the simulation results on case12 and case15 are more satisfactory than
those on case10. Similarly, the simulation accuracy of the OM-DBCM also gradually
decreased with the increasing of the ratio of coarse to fine grids. Overall, the benefit of
using the IM-DBCM for the flood simulations is evident.

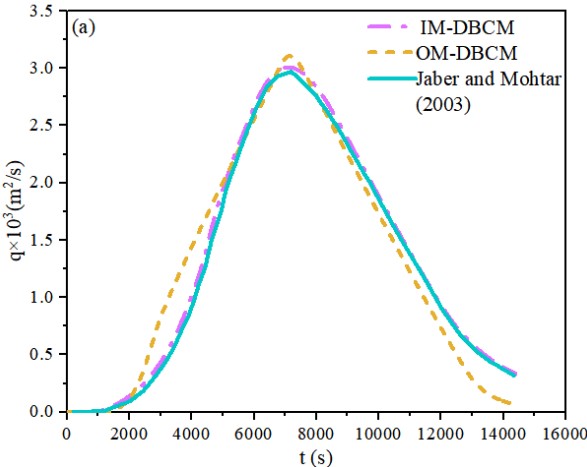






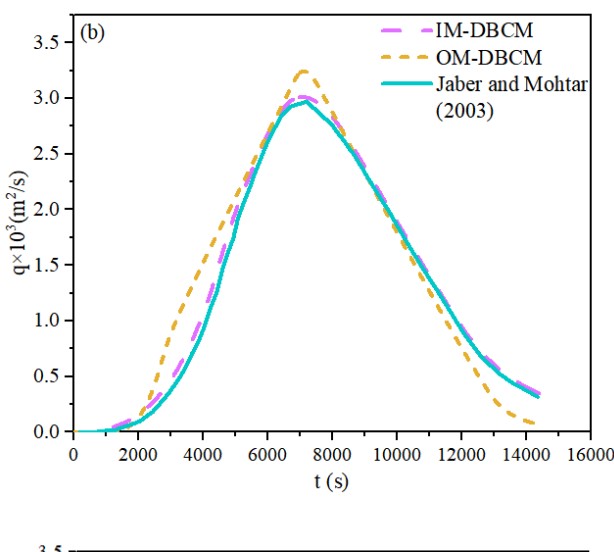


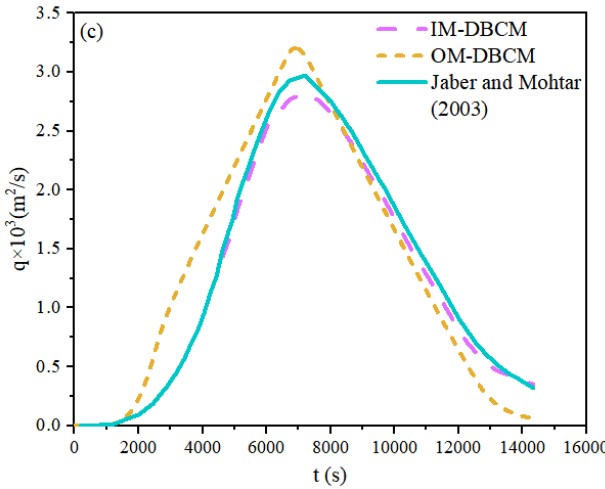


Figure 6 Hydrographs obtained from different models: (a) case12, (b) case15 and (c)

case10

**3.2 2D rainfall-runoff experiment**

In this case, the IM-DBCM was used to compute the hydrograph generated by

uniform rainfall conditions over a simple 2D geometry. The numerical results were

compared with experimental data obtained in a laboratory model developed by Cea et

al. (2008). The 2D geometry used in the experiment comprised a rectangular basin



composed of three stainless-steel planes, each with a slope of 0.05. The basin had two
walls that increased the residence time of the runoff in the basin and the length of the
outlet hydrograph. The geometric dimensions of the basin are shown in Figure 7.

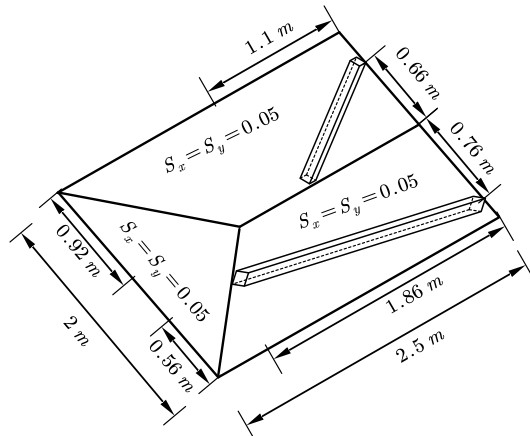


Figure 7. Geometry and size of the 2D basin for the rainfall-runoff experiment

Two rainfall intensities were simulated. In case01, the rainfall intensity was 317

mm/h for 45 s. In case02, the rainfall had an intensity of 320 mm/h for 25 s, then it
stopped for 7 s and started again continuing for 25 s with an intensity of 328 mm/h.

The computational basin was divided into coarse and fine grids based on the D∞

algorithm. The size of the fine grids was 0.01m × 0.01m, whereas that of the coarse
grids was 0.02m × 0.02m. The grid partition is presented in Figure S2 in Supplement.
According to Cea et al. (2008), the Manning coefficient was 0.009 $s/m^{1/3}$.

Figure 8 shows a comparison between the numerical and experimental outlet

hydrographs. The shape of hydrographs was well predicted in both cases, indicating
that the IM-DBCM could capture the flow process and exhibited satisfactory accuracy.
In case02, the first peak discharge rate occurred when the rainfall stopped for the first
time. Subsequently, the discharge rate began to decrease. After 7 s, rainfall started again,
and the discharge rate continued to decrease. The RMSEs of discharge simulated by



IM-DBCM in case01 and case02 were 0.107 and 0.023 respectively. The numerical
results were in good agreement with the experimental data. Compared to the results
obtained from OM-DBCM, the simulation results obtained from IM-DBCM were
closer to the experimental data. The results for case01 were slightly over-predicted by
the OM-DBCM.

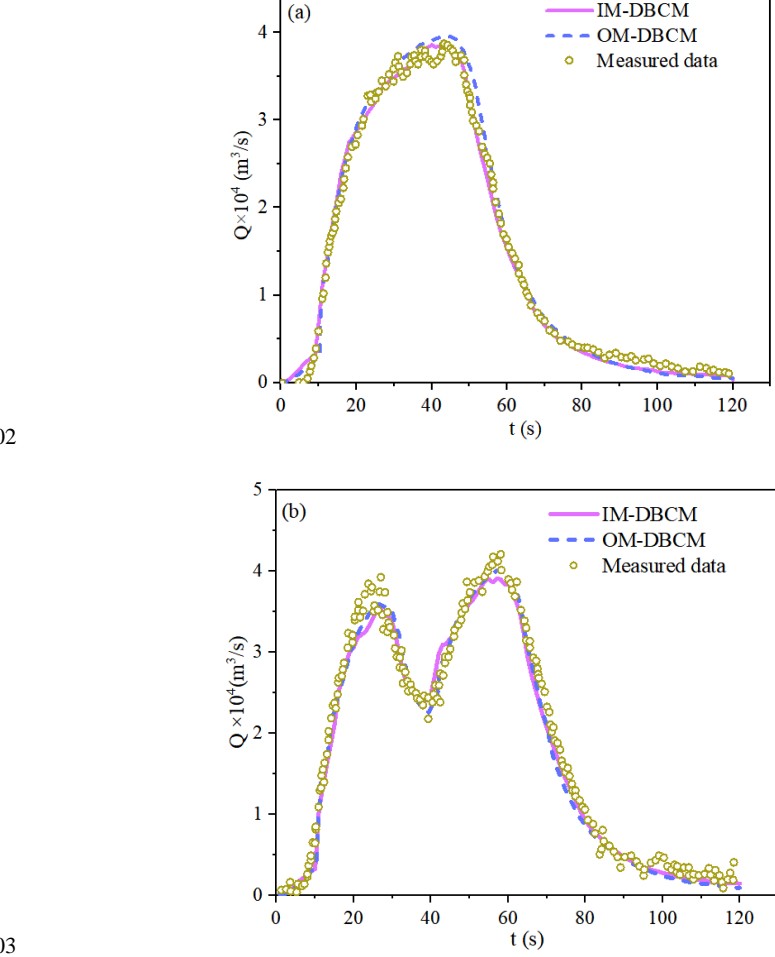



Figure 8. Simulated and measured discharge rate at different cases: (a) case01 and (b)

case02

To verify the conservation of the IM-DBCM, the inflow and outflow of different





cases in this basin were determined to represent the water balance, as shown in Figure
9. In case01, the outflow increased with the increasing of simulation time, whereas the
water storage increased first and then decreased. When the rainfall stopped at 45 s,
water was discharged from the basin; therefore, the water storage decreased. The sum
of the outflow and storage was equal to the accumulated rainfall, indicating that the IM-
DBCM can ensure the conservation of water mass. In case02, the outflow continuously
increased. Two peak flows were observed for the water storage, which was caused by
the intermittent rainfall. Overall, the sum of the outflow and water storage was equal to
the accumulated rainfall, indicating that the IM-DBCM ensured mass conservation.

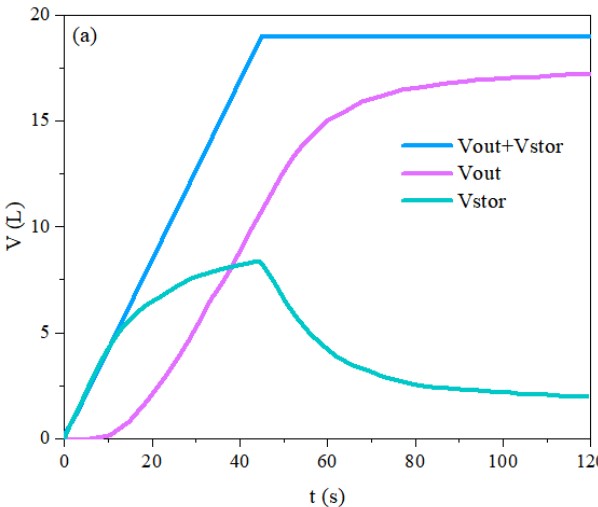






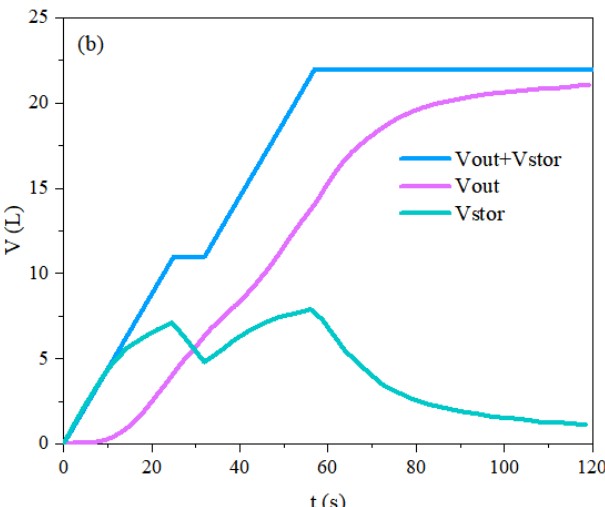


Figure 9. Inflow and outflow for different cases: (a) case01 and (b) case02, where

"Vout" refers to the outflow and "Vstor" refers to water storage in the computational

basin

**3.2 Flood simulation in a natural watershed**

The Goodwin Creek watershed, located in Panola County, Mississippi, USA, is

often selected as a benchmark to assess the capability of flood models because of

sufficient available observed data. Drainage is westerly to Long Creek which flows into

the Yocona River, one of the main rivers of the Yazoo River, a tributary of the

Mississippi River. The Goodwin Creek watershed covers an area of 21.3 km$^2$. The

overall terrain gradually decreased from northeast to southwest, which is consistent

with the trend of the main channel, and the elevation ranged from 71 to 128 m. The

computational basin and bed elevations are shown in Figure 10.

Land use in this watershed was divided into four classes including forest, water,

cultivated, and pasture, and their Manning coefficients were 0.05, 0.01, 0.03, and 0.04,

respectively (Sánchez, 2002). The infiltration coefficients of different soil types were

determined according to Blackmarr (1995). The rainfall event in sixteen rain gages (see



Figure 10) of October 17, 1981 was chosen for simulation (Sánchez, 2002), and the
inverse distance interpolation method (Barbulescu, 2016) was used to calculate the
precipitation over the entire watershed. The rainfall duration was 4.8 h. Rainfall was
spatially distributed at different times, as shown in Figure S3 in Supplement. There
were measured data in six observation stations (i.e., 1, 4, 6, 7, 8 and 14) Blackmarr
(1995), whose locations were shown in Table S1 in Supplement, and the simulated
results were compared with the measured data in these stations.

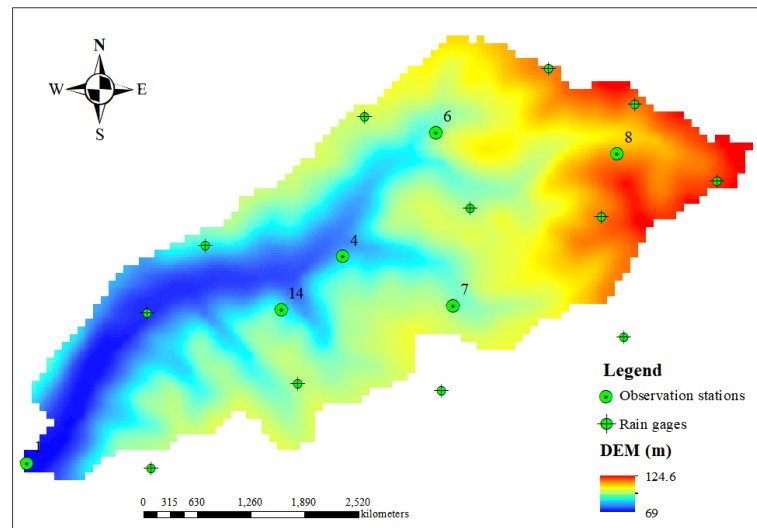


Figure 10. Overview of the Goodwin Creek watershed

The simulations were performed for 12 h. Different cases with various grid

resolutions were developed to verify the computational efficiency and numerical
accuracy of IM-DBCM, as listed in Table 2. In M-DBCM, the rivers were covered by
fine-grid cells with dimensions of 10 m × 10 m, whereas the coarseness in the rest of
the domain was increased to higher levels, as presented in Figure S4 in Supplement.

Table 2. Different cases designed to simulate the Goodwin Creek watershed

| Cases  | The ratio of coarse to fine grids | Number of grids |
|--------|-----------------------------------|-----------------|
| case12 | 1:2                               | 104,555         |



| case15 | 1:5 | 65,240 |
|---|---|---|
| case10 | 1:10 | 59,431 |

The OM-DBCM was also used to simulate the rainfall runoff with the same
resolutions. The Nash-Sutcliffe efficiency (NSE) was used to quantify errors in each
model. NSE ranges between $-\infty$ and 1.0, with NSE=1 being the optimal value. The
NSEs of IM-DBCM and OM-DBCM were shown in Table 3. From this table, the NSEs
of IM-DBCM were higher than that of OM-DBCM at most stations, which was
probably caused by the different interpolation method at the interface between coarse
and fine grids. It is verified that the IM-DBCM has relatively high accuracy in
simulating rainfall-runoff. In OM-DBCM, it is unreasonable to make the variables at
the interface between coarse and fine grids equal to that at the cell center, which can
bring errors. The induced error will increase as the ratio of coarse and fine grids increase.
Therefore, it is also observed that the NSEs of OM-DBCM decreased with the increased
ratio of coarse and fine grids. It is indicated that the ghost cells and bilinear interpolation
used in the IM-DBCM to interpolate variables between coarse and fine grids can make
the simulation more reasonable.
Table 3 NSEs of different models ("IM" and "OM" refer to IM-DBCM and OM-
DBCM, respectively)

| Station | G1 | | G4 | | G6 | | G7 | | G8 | | G14 | |
|---|---|---|---|---|---|---|---|---|---|---|---|---|
| Model | IM | OM | IM | OM | IM | OM | IM | OM | IM | OM | IM | OM |
| case12 | 0.9496 | 0.9108 | 0.9611 | 0.9011 | 0.9904 | 0.8982 | 0.9658 | 0.9004 | 0.9435 | 0.9104 | 0.9311 | 0.8804 |
| case15 | 0.9399 | 0.8766 | 0.9404 | 0.8800 | 0.9426 | 0.8819 | 0.9258 | 0.8931 | 0.9341 | 0.8942 | 0.9001 | 0.7942 |
| case10 | 0.9207 | 0.8261 | 0.8907 | 0.8435 | 0.9513 | 0.7977 | 0.9358 | 0.8525 | 0.9358 | 0.8678 | 0.9135 | 0.8078 |



Figure 11 shows a comparison of the measured and simulated hydrographs by IM-
DBCM at the monitoring gauges, whose locations are presented in Figure 10. At all
gauges, the hydrographs obtained from different cases were well aligned with the
measured data, which indicates that the IM-DBCM could reliably reproduce the flood
wave propagation in the complex topography. The results of case12, in general, were
better than those of case15 and case10, especially at station G1. A possible explanation
is that a finer grid is needed to better capture the watershed geometry and obtain more
satisfactory simulation accuracy. The cell size of case15 and case10 is larger than that
of case12.
Compared with other stations, at station G1, the simulation results obtained from
case15 and case10 deviated substantially from the measured data, especially at receding
limb of the hydrographs. We deduced that the reason for this discrepancy is not the
mesh partitioning, but the location of the G1. G1 is located at the watershed outlet,
where water flows out of the watershed from here. The errors generated upstream may
be accumulated at this station. Despite the deviation, the overall trend of the
hydrographs indicated that the IM-DBCM is satisfactory and can reliably reproduce
flood wave propagation in complex topography.

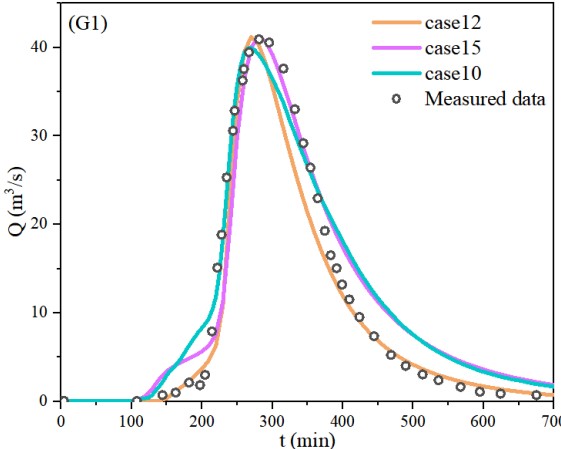




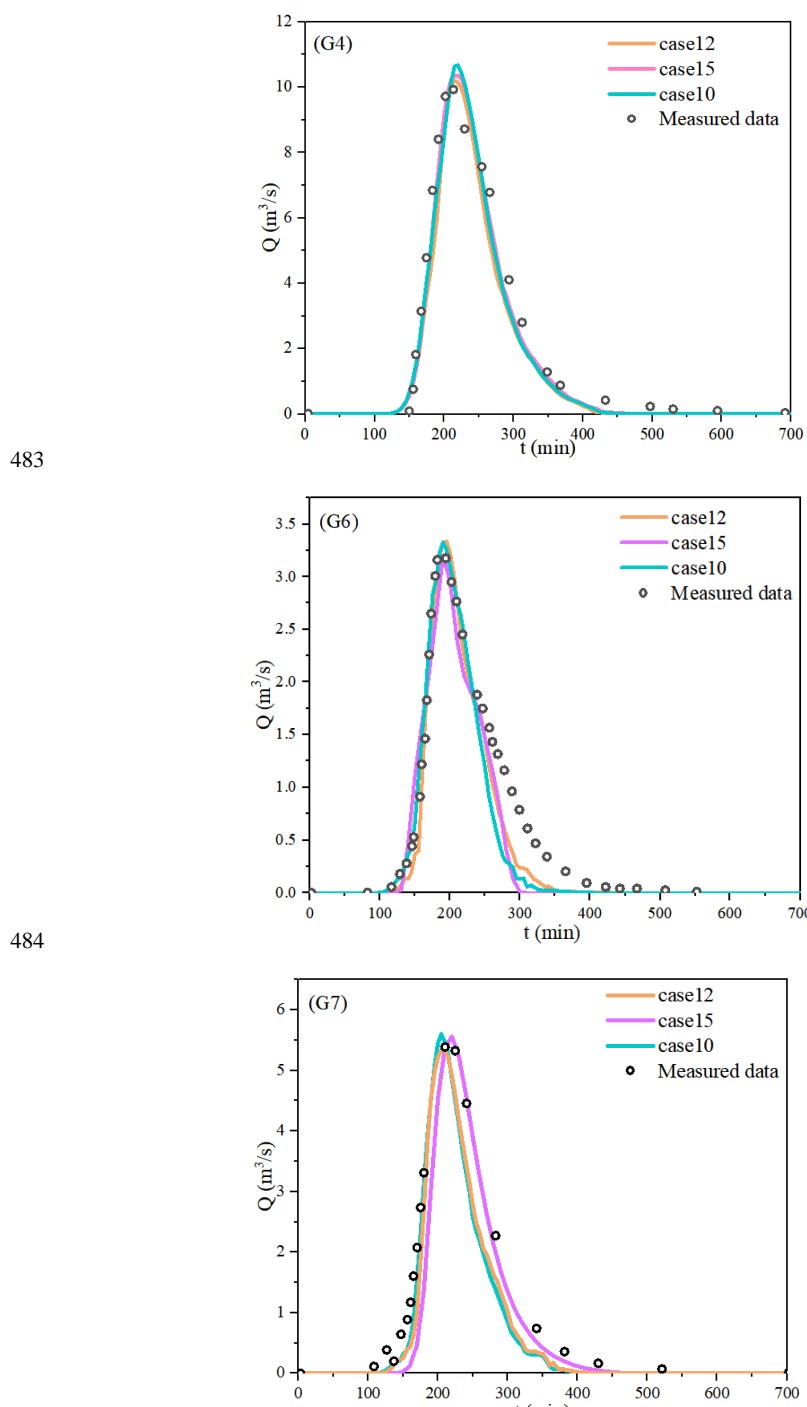








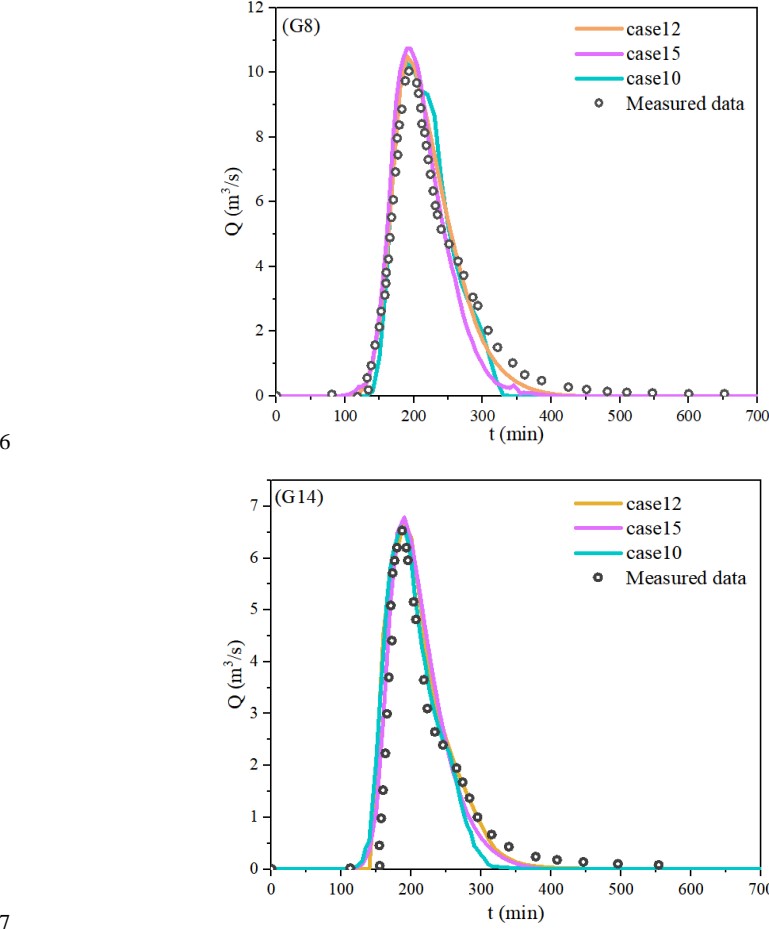


Figure 11. Hydrographs obtained from different cases

The water depth and location of the coupling interface at different times are shown

in Figure 12. The position of the coupling interface was time-dependent. From 0 to 5 h,
the water depth in the computational basin increased with the rainfall. Once the water
depth was higher than the predefined threshold, the regions were defined as inundation
regions and the hydrodynamic model was used to simulate the rainfall runoff. The water
depth peaked in the watershed at 5 h, as shown in Figure 12(a1), and most of the regions
were defined as inundation regions, as shown in Figure 12(a2). After 5 h, when rainfall
stopped, the water depth in the computational basin decreased (Figure 12(b1)). When





the water depth was lower than the predefined threshold, the inundation regions defined
last moment became non-inundation regions. Accordingly, as shown in Figure 12(b2),
the non-inundation regions expanded, whereas the inundation regions decreased. The
location of the coupling interface was shifted to the inundation regions defined at the
last moment. The results indicated that the coupling interface shifted during the
simulation, which was consistent with the flood migration process.

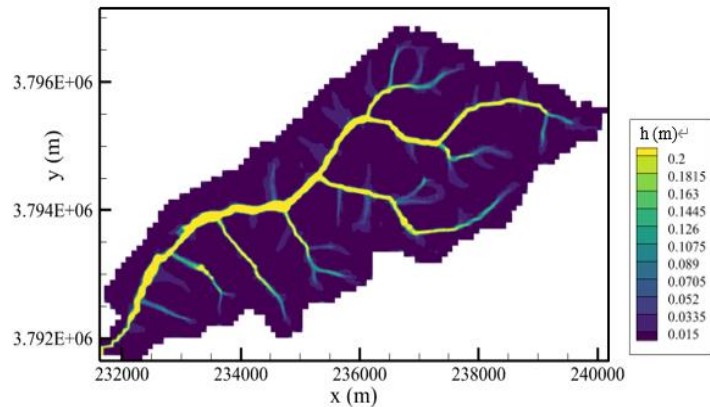


(a1) Water depth at $t = 5$ h

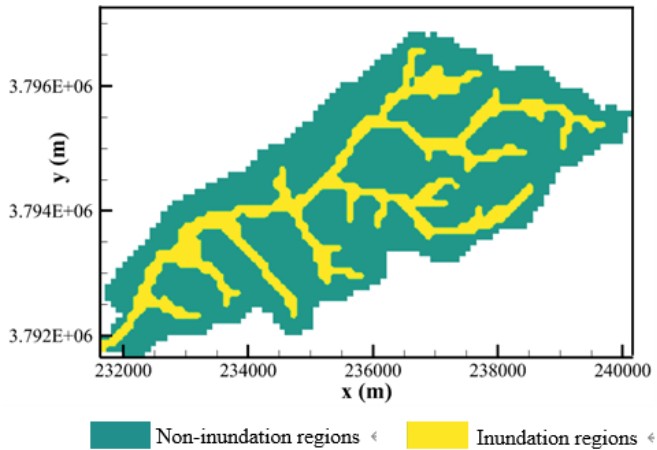


(a2) Position of the coupling interface at $t = 5$ h



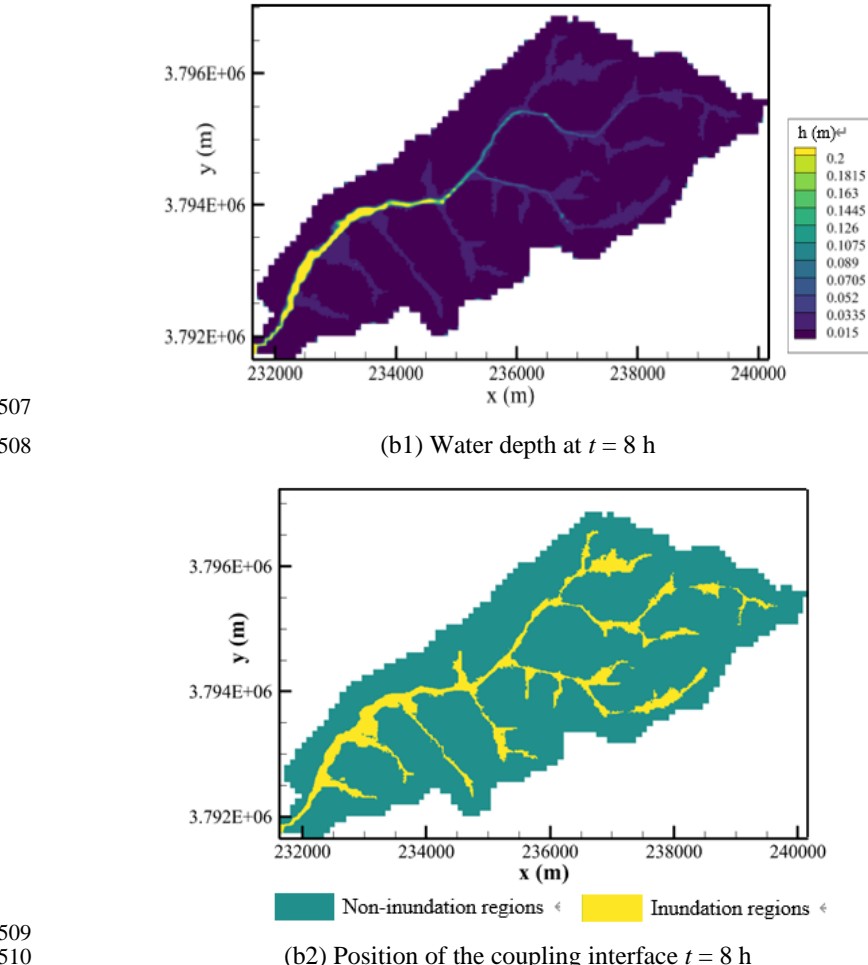


(b1) Water depth at *t* = 8 h

(b2) Position of the coupling interface *t* = 8 h

Figure 12. Water depth and position of the coupling interface of the hydrologic-

hydrodynamic model at different times

In terms of efficiency, the total execution time of IM-DBCM was compared with

the uniform grid-based model (case00), as shown in Figure 13. The total execution time
of the different cases ranked from highest to lowest is as follows: case00 > case12>
case15> case10. Uniform fine grids were used to divide the computing zones in case00,
and 207,198 computational grids were generated. Compared with case00, most of the
areas were discretized with coarse grids, and only a small part of the regions was



calculated based on fine grids in IM-DBCM; the computational grids of the multi-grid-
based model (Table 2) were considerably lower than that of case00. Furthermore,
case12 required less computational time than case15 and case10. The size of the fine
grid cell was the same (10 m × 10 m) in case12, case15 and case10. However, the size
of the coarse grid cell was twice that of the fine grids in case02, whereas the size of
coarse grid cell was five or ten times that of fine grids in case 15 or case10. Therefore,
fewer computational grid points were presented in case15 and case10, which required
less time for calculation, and the computational efficiency could be further improved.
This indicates that the computational time decreases when the size ratio of the coarse
grid to the fine grid increases. Thus, the advantages of using IM-DBCM based on multi-
grids for large-scale flood simulations are evident. The difference in total runtime
between the IM-DBCM and OM-DBCM is the time spent on grid partitioning. In the
OM-DBCM, the computational domain is divided manually, which is highly subjective,
and the computational time varied from person to person.
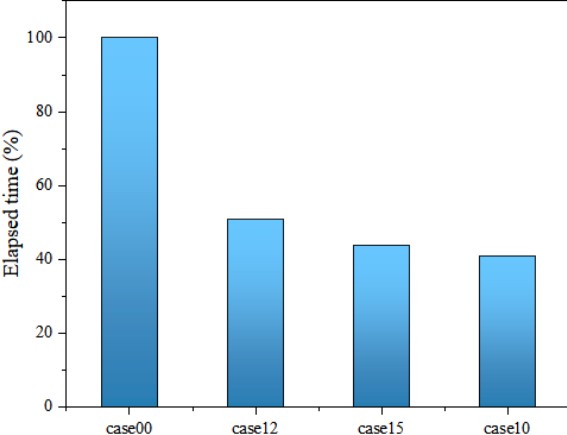

Figure 13 The relative difference in computation time of different cases

**4 Conclusions**

An improved dynamic bidirectional coupled hydrologic-hydrodynamic model



based on multi-grid (IM-DBCM) was presented in this study. A multi-grid system was
generated based on the D∞ algorithm, dividing regions that required high-resolution
representation using fine grids and the rest using coarse grids to reduce computational
load. A two-dimensional non-linear reservoir was adopted in the hydrologic model,
while two-dimensional shallow water equations were applied in the hydrodynamic
model. The hydrologic model was applied to the coarse-grid regions, whereas the
hydrologic and hydrodynamic models were coupled in a bidirectional manner for the
fine-grid areas. Different time steps were adopted in coarse and fine grids. Ghost cells
and bilinear interpolation were used to interpolate variables between coarse and fine
grids. The hydrologic and hydrodynamic models were dynamically and bidirectionally
coupled with a time-dependent and moving coupling interface.

The performance of IM-DBCM was verified using three cases. The IM-DBCM

was demonstrated to effectively simulate flow processes and ensure reliable simulation.
Compared with the OM-DBCM, the results obtained from the IM-DBCM were well
aligned with the measured data, and it could reliably reproduce the flood wave
propagation in complex topography. In addition to producing numerical results with
similar accuracy, the IM-DBCM saved computational time compared with the model
on fine grids. Furthermore, a moving coupling interface between the hydrologic and
hydrodynamic models was observed in the IM-DBCM. The IM-DBCM has both high
computational efficiency and numerical accuracy, which was adapted adequately to the
real-life flooding process and provided practical and reliable solutions for rapid flood
prediction and management, especially in large watersheds.

The IM-DBCM accurately and efficiently reproduces the flooding process and has

the potential for a wide range of practical applications. Adding a one-way
hydrodynamic model to the model could further enhance its performance. A one-way



model can simulate flow in a narrow river, saving more time than using a two-way
hydrodynamic model.
**Data availability**
Model simulation and calibration data are available upon request from the
corresponding author. Digital elevation model data are provided by the Geospatial Data
Cloud at http://www.gscloud.cn. The data sets of Soil Properties and Land cover are
provided by Sánchez (2002) and Blackmarr (1995). The rainfall and measured data
were Blackmarr (1995).
**Author contributions**
Yanxia Shen designed the methodology and carried out the investigation. Qi Zhou
provided the original model input data. The study was supervised by Chunbo Jiang.
Yanxia Shen prepared the first draft of the manuscript and Zhenduo Zhu revised and
improved the original manuscript.
**Competing interests**
The authors declare that they have no conflict of interest.
**Acknowledgements**
This study was supported by the National Natural Science Foundation of China
(Grant No. 52179068) and the Key Laboratory of Hydroscience and Engineering (Grant
No. 2021-KY-04). The authors thank the anonymous reviewers for their valuable
comments.

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
