# Peer review of "An improved dynamic bidirectional coupled hydrologic- hydrodynamic model for efficient flood inundation prediction Yanxia Shen, Zhenduo Zhu, Qi Zhou, Chunbo Jiang[*]"

_EGUsphere, 2023_

## Author Comment (AC1)

**Reviewer #1:**

First of all, sincerely thank you very much for your valuable comments. All your suggestions are very important and have important guiding significance for our writing and research. When revising the article, we considered thoughtfully what you have advised.

**1. Comment:** The paper presents an interesting approach of coupling hydrologic and hydrodynamic models to improve computational efficiency while maintaining numerical accuracy. However, to demonstrate the superiority of the proposed approach, it is essential to conduct a thorough comparison with state-of-the-art individual hydrology and hydrodynamic models. This will help highlight the advantages and necessity of the coupled modelling approach. It's crucial to show how the proposed method outperforms existing models in terms of both efficiency and accuracy.

In the introduction, the authors should focus more on recent progress in coupled hydrology-hydrodynamic models, especially with respect to their proposed coupling method, which seems different from the common coupling methods. Additionally, a detailed explanation of the non-uniform grid generation should be provided to give readers a better understanding of its significance in the proposed approach.

**Response to comment:** Thank you very much for your valuable comments. The coupling model can be divided into two types: external (one-way) and internal (two-way) coupling models (see Figure 1).

[Figure]

Figure 1 Classifications of coupled hydrologic and hydrodynamic models

One-way coupling model utilizes hydrographs obtained from hydrologic models as an input for hydrodynamic models, providing a one-way transition. Usually, the hydrologic model is run first and independently from the hydrodynamic model. External coupling models are powerful tools for watershed flood simulation, in particular large spatial and temporal scale, due to its convenience in model construction. However, the location of the boundary points limits the influence of the upland runoff to downstream waters. The runoff generation on both sides of the river is transferred to limited points upstream of the main stream or tributaries of the river network, resulting in an error in the peak flow rate of the boundary points. Since the flow information is transferred in one-way from hydrologic to hydrodynamic models, the external coupling cannot capture the mutual interaction between runoff production and flood inundation.

Moreover, mass conservation of water through the coupling interface cannot be guaranteed.

Two-way coupling models were further divided into four types: the coupling of hydrologic and 1D hydrodynamic models; the indirect coupling of hydrologic and 2D hydrodynamic models; full 2D hydrodynamic models, and the DBCM and M-DCBM proposed by our team. The characteristics and applications of different coupling models for flood simulation were detailed as follows.

(1) The hydrologic and 1D hydrodynamic models are calculated synchronously in time in the coupled hydrologic and 1D hydrodynamic models. The flow discharge rate obtained from the hydrologic model is treated as mass source of the 1D hydrodynamic model, while the water depth calculated in 1D hydrodynamic model is fed back to hydrologic model. The coupling of the Mike SHE and Mike11 is a typical example of coupling of hydrologic and 1D hydrodynamic models. The coupling of hydrologic and 1D hydrodynamic models lacks ability to accurately simulate flood inundation process in 2D regions, such as lakes, reservoirs, complex flows and estuaries where 2D or 3D computations are required.

(2) In order to overcome the lack of 2D hydrodynamic simulation in type-1, the coupling of hydrologic, 1D and 2D hydrodynamic models is proposed. In this coupling type, the runoff first flows into 1D rivers, and then discharge into the 2D inundation regions, such as lakes or reservoirs. The hydrologic model was coupled with 1D hydrodynamic model, and the 1D hydrodynamic model was coupled with 2D hydrodynamic model. This coupling type is an indirect coupling of hydrologic and 2D hydrodynamic models. For instance, Mike SHE and Mike11 are coupled to form Mike Urban, and Mike11 and Mike21 are dynamically coupled to form Mike Flood. The indirect coupling of hydrologic and 2D hydrodynamic models applied to simulate rainfall-runoff have been reported in many kinds of literature (http://doi.org/10.2166/wst.2017.504;http://doi.org/10.1016/j.jhydrol.2018.07.069; http://doi.org/10.1061/(ASCE)HY.1943-7900.0000485). Compared with type-1, this coupling type has satisfactory and acceptable accuracy and is widely used. However, in these models, the hydrologic model is not directly linked with 2D hydrodynamic model, which is inconsistent with the natural flood processes. In reality, runoff from the uplands may be simultaneously discharged into both 1D channel and 2D inundations, and the hydrologic and 2D hydrodynamic models should be linked directly. Direct dynamic coupling of hydrologic and 2D hydrodynamic model can reflect the flood process more truly. The dynamic bidirectional coupling of the hydrologic and the local 2D hydrodynamic models has been paid much attention.

(3) In this coupling type, only the runoff generation is calculated by the hydrologic model and considered as source term of the continuity equation of 2D hydrodynamic model, and then both the overland flow migration and inundation processes are all calculated by 2D hydrodynamic model. This coupling type is also called full 2D hydrodynamic model (HM2D). The HM2D can be used to simulate the complex flow patterns and achieve satisfactory results. In HEC-RAS (version 6.4), the flood process in 1D rivers was calculated using 1D hydrodynamic model, whereas the 2D diffusion wave equations (DWE) or shallow water equations (SWE) were solved in 2D regions.

The 1D hydrodynamic model was coupled with the 2D DWE or SWE. The HEC-RAS was also considered as the HM2D, since the 2D DWE or SWE were solved in the entire 2D regions. As the 2D hydrodynamic equations need to be solved in the entire watershed, the HM2D are still computationally prohibitive for large-scale applications, especially in regions where high-resolution representation of complicated topographic features are necessary. Therefore, the HM2D is typically applied to small and medium-sized watershed.

(4) The DBCM joins the hydrologic and hydrodynamic models into a single modelling framework by combing their code, where the governing equations of hydrologic and hydrodynamic models are reformulated and synchronously solved in a single code base. The information exchange between both portions of the code is performed internally within the same source code and does not involve the exchange of external input and output files. The hydrologic and 2D hydrodynamic model are coupled by a coupling moving interface (CMI), and the inundation and non-inundation regions change with the accumulation of rainfall, which is more consistent with the natural flood process. The DBCM framework was presented in the paper (http://doi.org/10.5194/nhes-21-497-2021). The classification, performance, applications and challenges of different coupling models were detailed by Shen and Jiang (2023). If you want to learn more about this, you can review it in https://doi.org/10.1007/s11069-023-06047-1.

To further improve the computational efficiency, we developed the M-DBCM (https://doi.org/10.3390/w13233454 ). In the original M-DBCM, the multi-grids were used to divide the computational domain, and the task consists of the following steps:

First, the areas prone to flooding disasters was identified based on two methods. A hydrologic model was used to simulate the flood disasters based on the coarse grids to determine the areas prone to flood disasters. Besides, the areas prone to flood disasters was also identified based on experience. Second, the areas prone to flooding disasters were divided using finer grids, whereas the others were discretized using coarse grids. The grid generation methods were detailed in Shen et al. (2021), if you want to learn more about this, you can review it in Shen et al. (2021) (https://doi.org/10.3390/w13233454). One limitation is that the grids need to be generated manually, which is highly subjective and uncertain. Therefore, we revised and improved the grid generation method, which is presented in Section 2.1 of the manuscript.

**2. Comment:** The modelling performance is highly influenced by the underlying mesh generation. Even with advanced adaptive methods using meter-scale data, there can be uncertainties impacting the modelling performance. It is recommended that the authors perform an uncertainty analysis on the mesh generation process to understand its potential effects on the model results.

**Response to comment:** Thank you for reading this article carefully and making valuable suggestions. There were many mesh generation methods, such as adaptive mesh refinement, static non-uniform grids, and the modelling performance is highly influenced by the underlying mesh generation. In Section 3.2 of the manuscript, cases

with different ratios of coarse to fine grids were developed. The computational efficiency and accuracy of different grid generations were evaluated. In future works, we can combine different mesh generation methods, such as the adaptive mesh refinement with the M-DBCM to study the influence of the gird generation on the simulation results. Besides, sensitivity analysis will also be performed to discuss the impact of parameters (such as Manning coefficient, the grid generation, ratio of coarse to fine grids) on the simulation results. Thank you for reading this article carefully and making valuable suggestions, which have important guiding significance for our writing and scientific research work.

**3. Comment:** The paper mentions the use of a coarse grid. What is the resolution of a coarse grid? What is the ratio between find grid resolution and coarse grid resolution? Will the coarse grid resolution/ratio have a large impact on modelling performance? Understanding the impact of this coarse grid resolution/ratio on the modelling performance is crucial.

**Response to comment:** In the proposed M-DBCM, the size of a coarse grid is an integer multiple of that of a fine grid. The fine grids were first used to divide the areas prone to the flood disasters, and then the coarse girds were used to discretize other areas. As the size of the fine grids varies in different computational domain, the size of coarse grids is also different. In a computational domain, compared with the fine grids, a grid with a larger size is referred to as a coarse grid.

The computational accuracy and efficiency were influenced by the ratio of coarse to fine grids, which was presented in the Section 3.2. Besides, the influence of the ratio of coarse to fine grids on the computational efficiency was detailed by Shen and Jiang (2023). If you want to learn more about this, you can review it in this paper (http://doi.org/10.2166/hydro.2023.131)

**4. Comment:** Providing a detailed description of the hydrology and hydrodynamic components, especially on their treatment of flow variables (e.g., discharge, depth), would greatly enhance readers' understanding of the coupling process at the interface. This information is vital to evaluate the robustness of the proposed coupling approach.

**Response to comment:** In hydrologic model, a 2D nonlinear reservoir model, including water balance and Manning equations, was used to simulate rainfall-runoff, and the governing equations were listed from Eqs. (7) to (11) in the manuscript. The shallow water equations were solved in hydrodynamic model. Since the shallow water equations were commonly used in most models, we did not detail the hydrodynamic model taking into account the length of the manuscript.

The Finite volume method of conservation scheme was used to discretize the governing equations of hydrologic and hydrodynamic model. A Harten-Lax-van Leer contact (HLLC) approximate Riemann solver was used to calculate the fluxes through the cell interface (see Figure 2).

The governing equations of hydrologic and hydrodynamic models are discretized on structured grids (see Figure 2). The hydrologic model is rational for the continuous non-inundation regions, and hydrodynamic model is rational for the continuous

inundation regions. However, since discontinuity existed at the coupling moving interface (CMI), the single hydrologic or hydrodynamic models were not acceptable, which was a challenge for the model calculation. It is necessary to apply suitable numerical schemes to calculate the fluxes through the CMI.

[Figure]

Non-inundation regions **CMI** Inundation regions

Figure 2 Model calculation at inundation regions, non-inundation regions and CMI

A pair of characteristic waves was used to determine the fluxes calculation methods through the CMI. The characteristic waves were calculated as follows:

$$S_L = u_{i,j} - \sqrt{gh_{i,j}} \tag{1}$$

$$S_R = u_{i+1,j} - \sqrt{gh_{i+1,j}} \tag{2}$$

where $S_L$ and $S_R$ are the characteristic waves; $u$ is the flow velocity (m/s); $h$ is the water depth (m); subscript $(i, j)$ and $(i+1, j)$ refer to the cells in non-inundation and inundation regions, respectively.

If $S_R > 0$ and $S_L > 0$, the fluxes through the CMI were calculated by the hydrologic model, and the CMI may move toward the non-inundation regions. Therefore, the non-inundation regions shrunk, whereas the inundation regions expanded. Only mass conservation through the CMI can be considered in this situation.

If $S_L < 0 < S_R$, the fluxes were calculated by both hydrologic and hydrodynamic models, and the CMI remained unchanged.

If $S_L < 0$ and $S_R < 0$, the fluxes are calculated by the hydrodynamic model, and the CMI may move toward inundation regions. Therefore, the inundation regions shrunk, whereas the non-inundation regions expanded. Both the mass and momentum conservation through the coupling boundary were obtained in the latter two situations. The couplings were detailed in Jiang et al. (2021) (http://doi.org/10.5194/nhes-21-497-2021) and Shen et al. (2021) (https://doi.org/10.3390/w13233454).

**5. Comment:** The paper uses small test cases to evaluate the modelling efficiency. However, it is important to validate the model's performance on larger scales, to ensure its practical applicability. Even the hydrodynamic models working on ~10m-30m can be configured for model run covering an area of several hundred kilometers and quite efficiently.

**Response to comment:** The proposed model in the manuscript has high computational

efficiency compared with full 2D hydrodynamic model. The computational domain was divided using grids with different sizes. The areas prone to flood disaster were divided into fine grids, while other areas were discretized into coarse grids. The hydrologic model was applied to coarse grids, whereas the hydrodynamic model was only solved in local fine grids. Different time steps were accepted in coarse and fine grids. However, the uniform fine grids were used to divide the computational domain in full 2D hydrodynamic model, and the 2D hydrodynamic model was solved in entire computational domain. The performance of the M-DBCM was evaluated by Shen and Jiang (2023). If you want to learn more about this, you can review it in http://doi.org/10.2166/hydro.2023.131

Shen and Jiang (2023) also showed that the larger the computational domain, the more pronounced the improvement in computational efficiency of the model. However, the main drawback of the proposed M-DBCM is the applications, due to the difficulty of the data collection including input data, observation data. In future works, we will apply the proposed M-DCBM to watersheds of different sizes. We sincerely hope we have the opportunity to continue our cooperation and publish our study in this journal. Thank you for reading this article carefully and making valuable suggestions, which have important guiding significance for our writing and scientific research work.

**6. Comment:** The choice of using Fortran for coupling the two modelling components, while the SWMM model is written in C++, raises questions about the rationale behind this decision. The authors should provide a clear explanation for this choice, considering factors like compatibility, performance, and ease of implementation.

**Response to comment:** C++ and Fortran are widely used in scientific research. There were many differences between C++ and Fortran. C++ is widely used in various domains such as system-level programming, game development, and graphical user interface development. Its flexibility and performance make it a versatile programming language. Fortran is primarily used in scientific computing, numerical simulation, and engineering calculations. It has rich libraries and optimization tools specifically designed for mathematical and scientific computations.

Our team started developing the coupled hydrologic-hydrodynamic model five years ago, when we have a software solving the hydrodynamic model based on Fortran language. It is convenient to develop the coupled hydrologic-hydrodynamic model based on the existing code. Therefore, we developed the coupled model based on Fortran language. We still use Fortran language to ensure continuity in the work of developing model.

However, since the C++ has more advantages than Fortran and is more widely used, we will develop the coupled model based on C++ in future works. It is thus more convenient to discuss the proposed model with other researchers. Thank you for reading this article carefully and making valuable suggestions, which have important guiding significance for our writing and scientific research work.

---

## Author Comment (AC2)

Dear reviewer,

Thank you very much for your letter and for the considerable comments. Those comments are all valuable and helpful for revising and improving our paper, as well as important guiding significance to our research.

I want to discuss the first comments that the respected reviewers proposed. The Mike series models are the most mature and widely used models, and the HEC-RAS was a new modified model in 2023 Therefore, I want to compare these two models with the DBCM proposed by our team.

(1) The Mike series models are the most mature and widely used models, which is the indirect coupling of hydrologic and 2D hydrodynamic models. For instance, Mike SHE and Mike11 are coupled to form Mike Urban, and Mike11 and Mike21 are dynamically coupled to form Mike Flood. The indirect coupling between the hydrologic and the 2D hydrodynamic models can be developed by coupling Mike Urban and Mike Flood. The 1D hydrodynamic model is a connection channel between the hydrologic and the 2D hydrodynamic models.

Compared with the coupling of hydrologic and 1D hydrodynamic models, this coupling type has satisfactory and acceptable accuracy and is widely used. As the 2D hydrodynamic model is only calculated in local inundation regions, its computational efficiency is greatly improved in comparison with the full 2D hydrodynamic model. However, in this coupling type, it is assumed that the water first discharges into the 1D rivers, and then flows through 1D rivers to the 2D regions. The hydrologic model is not directly coupled with the 2D hydrodynamic model, which is inconsistent with the actual flood processes. In reality, water may be discharged into both 1D channel and 2D waterbodies simultaneously, and the hydrologic, 1D and 2D hydrodynamic models should be linked directly. Direct coupling of hydrologic and 2D hydrodynamic models can reflect the flood processes more truly, which deserves more attention.

(2) HEC-RAS (version 6.4) was revised and improved in 2023. Figure 1, from the HEC-RAS 2D User's Manual, Version 6.4, Exported - July 2023, shows the multiple 2D inundation regions for floodplains that are connected with the 1D river channels. In HEC-RAS, the flooding process in 1D rivers is simulated by a 1D hydrodynamic model, whereas the flooding process in 2D regions is simulated using 2D diffusion wave equations (DWEs) or 2D shallow water equations (SWEs). The 1D hydrodynamic model is coupled with the 2D DWEs or SWEs. If the 2D regions are discretized into finer grids and the flooding process is simulated using 2D SWEs, the 1D hydrodynamic model is coupled with the 2D SWEs. In this way, the HEC-RAS is similar to Mike Flood. It has high numerical accuracy but is computationally prohibitive for large-scale applications. Conversely, if the 2D regions are discretized into coarse grids and the flooding process is simulated using 2D SWEs, the 1D hydrodynamic model is coupled with the 2D DWEs. In this way, the HEC-RAS is similar to the coupled Mike SHE and Mike 11, which can expand the application scale at the cost of reducing the accuracy.

HEC-RAS has the ability to have any number (within the computer's memory limitations) of separat
2D flow areas within the same geometry file. Multiple 2D flow areas can be added in the same way
storage areas. Hydraulic connections can be made from 2D flow areas to 1D Elements, as well as
between 2D flow areas. See the example in Figure 3-42.

[Figure]

Figure 1 The computational domain of the HEC-RAS obtained from HEC-RAS 2D
User's Manual Version 6.4 Exported - July 2023

(3) In the DBCM or M-DBCM proposed by our team, the computational domain
is divided into non-inundation and inundation regions, and the area of non-inundation
regions is much larger than that of inundation regions. The hydrologic model is applied
to non-inundation areas, whereas the 2D hydrodynamic model is applied to the local
inundation areas. When the rain intensity increased, the inundation regions expanded
because of the gradual accumulation of surface water volume. The inflow discharge
positions, flow path, and discharge values subsequently changed. Therefore, a coupling
moving interface (CMI) is formed between the inundation and non-inundation regions.
The hydrologic and 2D hydrodynamic models are coupled via this CMI. In DBCM, the
results of the hydrologic model affect the 2D hydrodynamic computation, and the
results of the hydrodynamic model also affect the hydrologic computation, which can
take into account effects due to overflowing in the floodplain, backwater effects at the
confluences.

To further improve computational efficiency, multi-grids were used to divide the
computational domain. The areas prone to flood disasters were divided into finer grids,
while the others were divided into coarse grids. The hydrologic model was applied to
coarse grids, while the hydrodynamic model was applied to fine grids. Different time
steps were accepted in fine and coarse grids.

Compared with the Mike series model, the coupling mechanism of DBCM is more
consistent with the natural flood disaster. Compared with HEC-RAS, it can save
computation time and has better numerical stability.

The DBCM has potential development if further improvement is made. This model
can be further improved by adding a 1D hydrodynamic model to it. The flow in a narrow

river can be simulated using a 1D model. And the direct coupling of hydrologic, 1D and 2D hydrodynamic models will be proposed in future works.

Chunbo Jiang
jcb@tsinghua.edu.cn

---

## Author Comment (AC3)

**Reviewer #2:**

First of all, sincerely thank you very much for your valuable comments. All your suggestions are very important and have important guiding significance for our writing and research. When revising the article, we considered thoughtfully what you have advised.

**1. Comment:** The authors present an improvement to the multi-grid hydrological/hydrodynamic SWMM/IM-DBCM model which partitions the model domain into a coarse resolution part (away from rivers) and a fine resolution part (areas susceptible to inundation). The authors present the mesh-generation approach and test the model in 4 configurations with a variable number of grid cells (ranging between 59k-207k cells). Results of discharge are compared to observed values, inundation depth is presented without comparison to observations. Advances are said to stem from improved computational efficiency, the main reason for the multi-grid approach of the model, while retaining an acceptable model performance.

**Response to comment:** Thank you very much for your valuable comments. SWMM is a direct one-way coupling of semi-distributed hydrological and 1D hydrodynamic models. Since the 1D nonlinear reservoir method is used to simulate the runoff routine, it is difficult to directly coupled with 2D hydrodynamic model. The M-DBCM proposed by our team is the direct dynamic two-way coupling of distributed hydrological and 2D hydrodynamic model.

In the section 3.2, flood process in natural watershed was simulated using the improved M-DBCM. The simulation data was collected from the references, Yu and Duan (2012); Sánchez (2002); Blackmarr (1995). Only the discharge hydrographs were obtained in the observation stations, therefore, we have compared the simulated discharge hydrographs with the measured data to evaluate the performance of the proposed model.

In the proposed model, the inundation and non-inundation regions were changed with the water depth; the hydrologic and hydrodynamic models were coupled through the moving interfaces. Therefore, we have presented the water depth and positions of coupling interfaces in Figure 12 to show the changing process of inundation and non-inundation regions with water depth.

However, our current research work lacks data, especially the measured data used to evaluate the performance of proposed model. In future works, we will collect more data, such as the water depth and discharge to further evaluate the performance of proposed model.

Ref:
[1] Yu C. and Duan J. G., Simulation of Surface Runoff Using Hydrodynamic Model, Journal of Hydrologic Engineering (ASCE), 2017, 22(6): 04017006

[2] Sánchez, R. R., GIS-based upland erosion modeling, geovisualization and grid size effects on erosion simulations with CASC2DSED. Thesis (Ph. D.) --Colorado State University, 2002.

[3] Blackmarr, W., Documentation of hydrologic, geomorphic, and sediment transport measurements on the Goodwin Creek experimental watershed, northern Mississippi, for the period 1982–1993, Res. Rep. 3, Agricultural Research Service, U.S. Dept. of Agriculture, Oxford, MS, 1995.

**2. Comment:** The state-of-the-art approach to the problem of variable resolution grids is a coarse-resolution hydrological model coupled with a 1D river routing model that activates a 2D model when channel capacity is exceeded. While the authors acknowledge this in the manuscript, they fail to compare their approach to results of such a model chain to demonstrate their advance. Admittedly, producing the same results with such a model chain to use as a baseline is a non-trivial task, but some comparison if not in the same catchment, should be considered mandatory.

**Response to comment:** Thank you for reading this article carefully and making valuable suggestions. The coupling model can be divided into two types: external (one-way) and internal (two-way) coupling models (see Figure 1). And the internal coupling model can be further divided into four types, as shown in Figure 2.

The coupling of the Mike SHE and Mike11 is a typical example of the coupling of hydrologic and 1D hydrodynamic models, as shown in Figure 2(a). The application of 1D modeling of overland flow is limited when developing precise and reliable flood maps in 2D inundation regions.

To overcome the lack of 2D hydrodynamic simulation in type-1, the coupling of hydrologic, 1D, and 2D hydrodynamic models is proposed. In this coupling type, the runoff flows into the 1D river or pipes first, and the hydrologic model is coupled with the 1D hydrodynamic model. And then, the water in 1D rivers or pipes can overflow into low-lying areas, the 1D and 2D hydrodynamic models are coupled in a two-way manner. This coupling type is an indirect coupling of hydrologic and 2D hydrodynamic models, as shown in Figure 2(b). For instance, Mike SHE and Mike11 are coupled to form Mike Urban, and Mike11 and Mike21 are dynamically coupled to form Mike Flood. The indirect coupling between the hydrologic and 2D hydrodynamic models can be developed by coupling Mike Urban and Mike Flood. The 1D hydrodynamic model is a connection channel between the hydrologic and the 2D hydrodynamic models.

In the Type-3, both the overland flow migration and inundation processes are all calculated by 2D hydrodynamic model, and the runoff generation is considered as the source term of the continuity equation of 2D hydrodynamic model. The type-3 has high numerical accuracy but low computational efficiency.

[Figure]

Figure 1 Classifications of coupled hydrologic and hydrodynamic models

[Figure]

Figure 2 Classifications of internal coupling models

It is observed that existing coupling models can not realize the dynamic two-way coupling of hydrology and 2D hydrodynamic models. The 1D hydrodynamic model was used to link the hydrologic and 2D hydrodynamic models. The runoff flows into the 1D river or pipes first, and the hydrologic model is coupled with the 1D hydrodynamic model. And then, the water in 1D rivers or pipes can overflow into low-lying areas; conversely, the water in low-lying areas can flow to 1D regions in return. The 1D and 2D hydrodynamic models are coupled in a two-way manner. This coupling type is an indirect coupling of hydrologic and 2D hydrodynamic models. In reality, however, water may be discharged into both 1D channel and 2D waterbodies simultaneously, and the hydrologic, 1D, and 2D hydrodynamic models should be linked

directly. Direct coupling of hydrologic and 2D hydrodynamic models can reflect the flood processes more truly, which deserves more attention.

Aiming to this problem, we have proposed a coupled hydrologic and 2D hydrodynamic models. In the proposed model, the 1D river channel and 2D inundation regions were not distinguished, and the 2D hydrodynamic model was applied to both regions.

There have been many published papers about the coupled 1D and 2D hydrodynamic models, and our future works may focus on the adding a 1D hydrodynamic model to the proposed M-DBCM, where the hydrologic model is used to simulate the runoff routing, the 1D hydrodynamic model is used to simulate the flood process in rivers and the 2D hydrodynamic model is used to reflect the inundation process in the low-lying inundation regions. Three coupling strategies, i.e., the coupled hydrologic-1D hydrodynamic module, coupled 1D-2D hydrodynamic module and coupled hydrologic-2D hydrodynamic module, are proposed.

However, the direct dynamic bidirectional coupling of distributed hydrologic and 2D hydrodynamic models is the key and important technology to develop the flood simulation models, and it is also the innovation of the M-DBCM. Besides, the multi-grids are used to divide the watershed, and the model can improve computational efficiency while maintaining numerical accuracy, which is the main difference between the proposed model and other existing models.

**3. Comment:** The authors present model runtimes of the four grid configurations in Fig 13. It is apparent that runtimes scale linearly when comparing the uniform grid case to the variable-grid case (cf case00 to case12). This relationship does not hold, however, in coarser configurations of the multi-grid models (cases 15/10), i.e., the runtime/grid is significantly higher. e.g., going from 105k cells to 59k cells only brings moderate efficiency savings of 10-15%. This shows the limits of the approach, presumably because more time is spent on coupling the coarse and fine grids.

**Response to comment:** Thank you for your valuable suggestions. We have proposed a parameter to quantitatively evaluate the computational efficiency of the M-DBCM (Shen and Jiang, 2023, http://doi.org/10.2166/hydro.2023.131). We defined the evaluation parameter as the ratio of the simulation time of the M-DBCM to that of the full 2D hydrodynamic model (HM2D), as shown in Eq. (1):

$$C = \frac{(1+\alpha_0)t_1 + t_2}{t_0} = \frac{(1+\alpha_0)\alpha \frac{A_1}{\Delta x_1^2}\left(\frac{T_{end}}{\Delta t_1}\right) + \beta \frac{A-A_1}{\Delta x_2^2}\left(\frac{T_{end}}{\Delta t_2}\right)}{\alpha \frac{A}{\Delta x_1^2}\left(\frac{T_{end}}{\Delta t_1}\right)} \tag{1}$$

where $C$ is the assessment parameter to evaluate computational efficiency of M-DBCM;

$t_1$, $t_2$ are the computation time on fine and coarse grids, respectively (s); $t_0$ is the computation time of HM2D (s); $\Delta x_1$, $\Delta x_2$ are the size of fine and coarse grids (m); $\Delta t_1$, $\Delta t_2$ are the time step on fine and coarse girds (s); $A_1$, $A_2$ are the area of coarse and fine grids, respectively; $T_{end}$ is simulation time (s); $\alpha$ and $\beta$ are the runtime of hydrodynamic and hydrologic models at one calculation node (s), which is depended on computer power and numerical model complexity. Since the hydrodynamic model is expressed by nonlinear hyperbolic equation and hydrologic model is expressed by linear equation, the calculation of the hydrodynamic model is more complicated than that of the hydrologic model, which results $\dfrac{\beta}{\alpha} < 1$.

The time step ratio of coarse grids to fine grids is equal to the size ratio of coarse grids to fine grids, as follows:

$$\frac{\Delta t_2}{\Delta t_1} = \frac{\Delta x_2}{\Delta x_1} \tag{2}$$

Based on Eq. (2), Eq. (1) can be rewritten as:

$$C = \frac{A_1}{A} + \frac{\beta}{\alpha} \left( \frac{\Delta x_1}{\Delta x_2} \right)^3 \frac{A - A_1}{A} \tag{3}$$

Define $n = \dfrac{A_2}{A}$ $(0 < n \leq 1)$, $\Delta t_2 = k \Delta t_1$ $(k \geq 1)$, Eq. (3) becomes

$$C = (1 - n) + \frac{\beta}{\alpha} \frac{1}{k^3} n \tag{4}$$

From Eq. (4), the computational efficiency of M-DBCM is not only related to the size ratio of coarse to fine grids, but the area ratio of coarse grids to entire domain. If the area of coarse-grid regions are much greater than that of the fine-grid regions, that is, $n \to 1$, the assessment parameter becomes $C \propto \dfrac{\beta}{\alpha} \dfrac{1}{k^3}$. It is indicated that the computational efficiency of M-DBCM exponentially improves with the increasing of the size ratio, as shown in Figure 3(a). If the size of coarse grids is much more than that

of fine grids, that is, $k \to \infty$, the assessment parameter becomes $C \propto (1-n)$. It is stated that the computational efficiency of M-DBCM improves linearly with the increasing of the area ratio of the coarse grids to entire domain, as shown in Figure 3(b).

[Figure]

Figure 3 The relationship between the evaluation parameter and the $n$ and $k$:(a) the relationship between the evaluation parameter and $n$; (b) the relationship between the evaluation parameter and $k$

We compared the graphs of three functions: $y = 1/x$, $y = 1/x^2$, $y = 1/x^3$, as shown in Figure 4. From this figure, compared with $y = 1/x$, in the $y = 1/x^3$, the $y$ values decrease sharply as $x$ increases. It is indicated that the computational efficiency of M-DBCM exponentially improves with the increasing of the size ratio. From this figure, in the $y = 1/x^3$, when $x< 10$, $y$ values are highly variable; however, when $x> 10$, although the $y$ value is decreasing, it is decreasing slowly. It is indicated that a ratio of coarse to fine grid between 1 and 10 may be suitable. We came to this conclusion after carefully reading the reviewers' comments. Thank you again for your valuable comments!

[Figure]

Figure 4 The graphs of $y = 1/x$, $y = 1/x^2$, $y = 1/x^3$

In the original manuscript, different cases were used to divide the Goodwin watershed, as shown in Figure 5. In the case12, case15 and case10, the number of coarse and fine grids are shown in Table 1. The number of fine grids accounts for half of the

total number of grids in case12, while the number of fine grids is much greater than the number of coarse grids in case15 and case10. The calculation time of coarse and fine grids is also reported in Table 1. The size of fine grids is the same in all the cases. In case12, the size of coarse grid is twice that of the fine grid, while the size of the coarse grid is five times and ten times that of the fine grids in case15 and case10, respectively. The number of grid cells ranked from more to less is as follows: case12> case15> case10. It is well-known that the more grids mean longer computational time. Therefore, case12 cost more computation time compared with case15 and case10.

[Figure]

(a) case12                                      (b) case15

(c) case 10

Figure 5 Grid partition of different cases

Table 1 The computation time of grids with different sizes (s)

|  | Case12 | Case15 | Case10 |
|---|---|---|---|
| The number of fine grids | 42474 | 42474 | 42474 |
| The number of coarse grids | 42517 | 7425 | 2153 |
| Computation time for fine grids | 4910.1 | 4890.32 | 4761.88 |
| Computation time for coarse grids | 243.8 | 16.28 | 2.19 |
| Total runtimes | 6900 | 6206 | 5800 |

The runtimes of different cases are shown in Figure 6. The total execution time includes the time for coarse and fine grids and others, such as the input and output of data, coarse and fine grids interpolation, coupling of the hydrological and 2D hydrodynamic models, wet and dry grid judgment, and so on. It is observed that the runtime for coarse grids decreases rapidly in different cases, which is related to the number of the coarse grids. In case12, case15 and case10, the number of the coarse grids is 42517, 7425, and 2153, respectively. Therefore, the runtime for the coarse grids

decreased rapidly.

However, the number of fine grids is much greater than that of the coarse grids, especially in case15 and case10. The 2D hydrodynamic model was solved in the fine-grid regions. Therefore, it cost more computational time compared with coarse grids, as shown in Figure 6, where the bule bar is the time spent on coarse grids. In all the cases, due to the large amount of calculations involved in fine grids, the time spent on fine grids accounts for a significant proportion of the total execution time. Therefore, the total computational time of all cases does not differ significantly.

[Figure]

Figure 6 The runtime for coarse and fine grids and the total runtime

Besides, we have calculated an example to further evaluate the computational efficiency of the proposed model. In this example, the sketch of the case is shown in Figure 7. The length of the plane is 182.88 m. A Manning's coefficient of 0.025 s/m$^{1/3}$ is recommended. The bed slope is 0.016. The constant rainfall intensity is 50.8 mm/h. The rainfall duration is 1,800s and the total simulation time is performed for 3,600 s. Different cases with various grid size ratios were designed to evaluate the performance of the M-DBCM, as listed in Figure 8. In M-DBCM, the size of fine grids is 1.83 m while the rest of the domain is coarsened to levels higher.

[Figure]

Figure 7 The schematic description of the example

[Figure]

(a) Case 12

(b) Case 15

(c) Case 10

Figure 8 The different size ratios of coarse grids to fine grids

The total execution time of different cases is shown in Figure 9. It is observed that the computational time of the case15 and case10 has been significantly decreased compared with that of case12. In this example, the number of fine grids is less than that of coarse grids, as listed in Table 2. A large proportion of the time spent on coarse grids, especially in case12. With the increasing of the ratios of the coarse to fine grids, the number of coarse grids is significantly reduced. Therefore, the runtime for coarse grids decreased rapidly, and the total execution time decreased significantly. Compared with the Goodwin watershed, the fine grid regions occupy a small proportion in this example. Therefore, the total execution time decreased rapidly with the increased of the ratio of coarse to fine grids.

[Figure]

Figure 9 The runtime for coarse and fine grids and the total runtime

Table 2 The number of fine and coarse grids

|  | Case12 | Case15 | Case10 |
| --- | --- | --- | --- |
| The number of fine grids | 144 | 144 | 144 |
| The number of coarse grids | 1400 | 352 | 144 |

There were two reasons to explain the respected reviewer's question. On the one hand, there were large area of fine grid regions, and the number of fine grids was higher than that of coarse grids in all the cases. The 2D hydrodynamic model was solved in the fine-grid regions, which cost more computational time compared with coarse grids. Due to the large amount of calculations involved in fine grids, the time spent on fine grids accounts for a significant proportion of the total execution time. Therefore, the total computational time of all cases does not differ significantly. On the other hand, as the respected reviewer pointed, the 1D hydrodynamic model would be added into the M-DBCM. The 1D hydrodynamic model is used to simulate the flood process in 1D rivers and the 2D hydrodynamic model is only used to simulate the flood process in the low-lying inundation regions. The low-lying inundation regions account for a small proportion of the total watershed regions. Therefore, this will greatly shorten the calculation time.

In future works, we will choose many more appropriate watersheds to evaluate the model performance, where the proportion of low-lying inundation regions to the total watershed area can be further reduced. In addition, the 1D hydrodynamic model will be added to the proposed model.

**4. Comment:** -l.74 -85 unclear, needs proofreading
**Response to comment:** We have proofread the manuscript thoroughly, especially the lines from 74 to 85.

**5. Comment:** -l.117 Ghost cells need to be defined before
**Response to comment:** We have defined the ghost cells in the introduction.

**6. Comment:** - l.498 and l.501 "last moment" > in the last time step?
**Response to comment:** The "last moment" means "in the last time step". We have revised in the revised manuscript.

---

## Author Response (AR2)

Dear editors:

Thank you very much for your letter and for the respected reviewers' comments concerning our manuscript entitled "An improved dynamic bidirectional coupled hydrologic-hydrodynamic model for efficient flood inundation prediction" (ID: egusphere-2023-1106). Those comments that the respected editor proposed are all valuable and very helpful for revising and improving our paper, as well as important guiding significance to our research. We have studied comments carefully and have revised the article which we hope meet with approval. There were new lines and page numbers in the revised manuscript. All the changes were marked using red bold in the revised manuscript. We also responded point by point to the reviewers' comments as listed below, along with a clear indication of the revision. Hope these will make it more acceptable for publication.

**Reviewer #1:**

First of all, sincerely thank you very much for your valuable comments. All your suggestions are very important and have important guiding significance for our writing and research. When revising the manuscript, we considered thoughtfully what you have advised.

**1. Comment:** While the DBCM is currently not available as an open-source model package, a comparative analysis with widely used software is essential in the presentation of the model results. This will elucidate the advantages of the proposed coupling mechanism, which claims to align more consistently with natural flood disasters. Additionally, it is crucial to demonstrate the computational efficiency and accuracy gains compared with state-of-the-art models.

**Response to comment:** Thank you very much for your valuable comments. In the Introduction, the advantage, disadvantage and applicability of existing coupling models were detailed. And therefore, the benefit of the DBCM was highlight compared to other coupled models.

Considering the length of the manuscript, we have removed the case in Section 3.2 of the original manuscript, and added another example in the revised manuscript. In this example, the full 2D hydrodynamic model (HM2D) and coupled Mike SHE and Mike 11 were also used to simulate the surface flow to verify the performance of the proposed IM-DBCM. If you want to learn more about this, you can review it in the Section 3.2 of the revised manuscript, which were also remarked using red bold.

Besides, we have also compared the computational efficiency of the IM-DBCM with the HM2D (see Figure 11), which was shown in the Section 3.3 in the revised manuscript.

**2. Comment:** Section 2.3.1: In the context of the hydrological model, a comprehensive consideration of the water balance is paramount. It is imperative to elucidate the methodology employed for estimating evaporation, infiltration, and ground water dynamics. Also, how land cover and soil moisture texture impact the hydrological

processes.

**Response to comment:** Thank you for reading this article carefully and making valuable suggestions. In the hydrologic model applied to IM-DBCM, only the surface flow processes, i.e., runoff generation and routing, are calculated. The runoff generation was equal to the rainfall intensity minus infiltration and evaporation rate, and the infiltration rate was calculated using Green-Ampt.

In a flash flood, surface flow is the most important factor to be considered, interflow, and underground runoff can be neglected due to the short rainfall and calculation duration (Hou et al., 2018; Li et al., 2021). However, a complete hydrologic model should include surface flow, interflow, and underground runoff. Regarding the surface hydrologic model, the ignored interflow and underground runoff processes likely lead to underestimations in flood simulation, especially for long-term simulations. In future works, the interflow and underground runoff could be calculated in the hydrologic model.

The aforementioned issues are discussed in the Conclusions section of the revised manuscript (lines 626-630), which are also highlighted in red bold. Furthermore, we noted that only surface runoff was simulated in the hydrologic model (Section 2.3.1 of the revised manuscript, lines 289-291).

References:

Hou, J., Wang, R., Liang, Q., Li, Z., Huang, M.S., Hinkelmann, R. (2018). Efficient surface water flow simulation on static cartesian grid with local refinement according to key topographic features. Computers & Fluids, 176, 117-134 https://doi.org/10.1016/j.compfluid.2018.03.024

Li, Z., Chen, M.Y., Gao, S., Luo, X.Y., Gourley, J.J., Kirstetter, P., Yang, T.T., Kolar, R., McGovern, A., Wen, Y.X., Rao, B., Yami, T., Hong, Y. (2021). CREST-IMAP v1.0: a fully coupled hydrologic-hydraulic modeling framework dedicated to flood inundation mapping and prediction. Environmental Modelling and Software, 141(1), 105051. http://doi.org/10.1016/j.envsoft.2021.105051

**3. Comment:** Evaluation of discharge exchange at the coarse-fine resolution interface is crucial. Specifically, when surface runoff transitions from a coarse grid to a fine grid, potential instability issues should be thoroughly examined. Though interpolation method is applied, will the possible sharp increase in discharge at the coarse-fine grid interface cause instability?

**Response to comment:** We acknowledge and fully understand your concerns. In the IM-DBCM, the multi-grids were used to discrete the computational domain. It may reduce numerical accuracy at the coarse-fine grid interface, compared to the model using fine grids discrete computational domains. The accuracy and stability of interpolation formats have been discussed in existing literature. For example, the Holly-Preissmann scheme employs values of the dependent variable and its derivatives to achieve high-order spatial interpolation (Holly and Preissman, 1978; Holly and Cunge, 1979). This approach avoided interpolation errors that exceed the maximum or minimum values.

In aerodynamic and hydrodynamic simulations, the nonlinear partial differential

nature of the Navier-Stokes or shallow water equations often leads to flow discontinuity, such as shock waves, hydraulic jumps, or tsunami waves. These phenomena result in abrupt spatial variations in the flow state. Linear interpolation of variables between coarse and fine grids can smooth out discontinuous solutions, resulting in substantial errors.

However, in the IM-DBCM proposed in this study, two types of connecting interfaces are presented, which divide the computing domain into three parts (see Figure 3 in the revised manuscript). The first part represents the coarse-grid areas, where the hydrologic model is used to simulate rainfall-runoff. The other two parts are located in the fine-grid areas. The regions between VII and CMI are defined as intermediate transition zones, where the hydrologic model is used to simulate the flooding process. These transition zones facilitate the application of different time steps in different grid cell sizes to improve computational efficiency. On both sides of the interface between coarse and fine grids, the hydrologic model was used to simulate the flood process. In the hydrologic model applied to the IM-DBCM, the Manning equation is employed to simulate surface runoff processes. As a linear partial differential equation, the Manning equation lacks a nonlinear convection term. Consequently, the flow state undergoes relatively smooth changes without exhibiting discontinuous solutions. Linear interpolation is applied to interpolate variables between coarse and fine grids, with the interpolated values falling within the range defined by the maximum and minimum values of the interval. This interpolation ensures that the result lies between these bounds, precluding the occurrence of increased flow at the interface of coarse and fine grid transitions.

The bilinear interpolation scheme employed for the spatial discretization of variables has been preliminarily validated through numerical examples, demonstrating its numerical stability and validity. The reviewers' suggestions provide invaluable guidance for enhancing the quality of this paper. Future research directions include exploring the relationship between various spatial interpolation schemes and simulation errors for different types of partial differential equations.

We have clarified the performance of the bilinear interpolation applied to the IM-DBCM in the revised manuscript, from lines 268 to 277.

References:

Holly F M, Preissman A. Accurate Calculation of Transport in Two Dimensions. (1978). Journal of Hydraulics Division, 103:1259-1277.

Holly F M, Cunge J. Coupled Dynamic Streamflow-Temperature Models Discussion. (1979) Journal of Hydraulics Division, 104: 316-318.

[Figure]

Figure 3 in the revised manuscript. Schematic diagram of grid generation, where $i$ and $j$ are the coordinates of coarse grid; $x$ and $y$ are the coordinates of fine grid; VII is the Variable Interpolation Interface and CMI is the Coupling Moving Interface

**4. Comment:** Concerning the mesh generation controlled by the D∞ algorithm, there is a suspicion that the mesh might be limited to a static structured nonuniform format, controlled by only the DEM features without consideration of the flow characteristics. It would be beneficial to clarify whether an adaptive mesh generation approach is employed, allowing the mesh to dynamically adjust over time in response to changing flow conditions. Such clarification would contribute to a more nuanced understanding of the model's spatial discretization strategy.

**Response to comment:** Thank you very much for your valuable comments. Initially, Adaptive Mesh Refinement (AMR) was applied to the DBCM (Yu 2019), but has since been replaced with static mesh due to the high computational cost and complexity of the AMR. Static structured non-uniform grids were used to discretize the computational domain based on the D∞ algorithm. The grid generation can be considered as a model preprocess, which is the foundation of flood simulation and can influence both computational accuracy and efficiency. In the watershed, when heavy rainfall occurs, the extent of the inundation regions can be primarily determined. Our focus is on flood dynamics in inundation regions. Therefore, a static grid can meet the requirements. For example, the Mike and SWMM models also employ static grids to discretize 2D inundation regions and have gained widespread usage.

AMR is commonly employed in scenarios where flow characteristics exhibit abrupt variations, such as aerodynamic shock waves, hydraulic jumps, and tsunami waves. Capturing discontinuous solutions necessitates local grid refinement, with the location of refinement dynamically adapting to the position of the discontinuities. Consequently, AMR are indispensable. However, AMR needs to segment and merge the grid elements repeatedly during the calculation, which can be time-consuming and offset the calculation time saved by the optimized grid. Besides, the mesh generation and flood simulation were compiled in the same code base, which increased the computation cost and time. Static non-uniform grids have increasingly received attention in recent years, which simplified grid generation procedure compared with AMR.

Flow characteristic variations arising from abrupt geometric changes in the computational domain can be captured using static local refinement grids, provided that the extent of these changes is limited. This approach offers computational time savings. In flood simulations, inundation regions are typically situated in low-lying 2D regions. The outer boundary of the inundation regions can be determined using DEM or calculating by hydrologic models. For example, the Mike series model employs static grids to delineate the boundaries of 2D low-lying waterlogged areas (Thompson et al., 2004). Given that the extent of these areas can be estimated in advance, static grids offer enhanced computational efficiency. The impact of employing hydrodynamic or hydrological models on the overall calculation results is minimal due to the shallow water depths and relatively low flow velocities at the boundary between the inundation and non-inundation regions. Consequently, static multi-grids are employed in flood inundation calculations in the proposed model. This approach involves coarse and fine grid discretization, followed by interpolation. It has been demonstrated to yield satisfactory results.

However, the structured grids were used to divide the computational domain based on D∞ algorithm. But the finite volume numerical scheme can be implemented on both structured and unstructured grids. In the future works, maybe the non-structure grids can be used to divide the computational domain based on D∞ algorithm.

We have compared and analyzed the performance of static non-uniform grids and AMR in Introduction (from lines 98 to 119). We have also clarified the grid technology applied to the proposed model in the revised manuscript from lines 148 to 149.

References:

Yu, W.: Research on Coupling Model of Hydrological and Hydraulics Based on Adaptive Grid. Ph.D. Thesis, Tsinghua University, Beijing, China, 2019.

Thompson, J.R., SoRenson, H.R., Gavin, H., Refsgaard, A.: Application of the coupled MIKE SHE/MIKE 11 modelling system to a lowland wet grassland in southeast England. Journal of Hydrology, 293(1-4): 151-179. 2004. http://doi.org/10.1016/j.jhydrol.2004.01.017

**5. Comment:** Given that the model is currently confined to your research group, it is advisable to articulate a broader research purpose, both in the introduction and the method section. This will provide a more comprehensive understanding of the main contributions that the DBCM brings to the field of flood modelling.

**Response to comment:** Many thanks for your suggestions. In the first paragraph of the introduction, we have explained the necessity of developing the coupling models. And next, we introduced the shortcomings of existing coupling models. Finally, we present the purpose and importance of the model we intend to develop. We aim to develop a model that can improve the computational efficiency while maintaining simulation accuracy. In the Conclusions, we have verified that the performance of the proposed model, which can be used for rapid flood prediction and management.

However, the proposed model has a relatively short timeframe, typically spanning two to three years. The widespread promotion and application, like Mike series, necessitate a protracted timeframe. Our ultimate objective is to disseminate our research

findings through scientific publications, thereby broadening the accessibility and comprehension of the proposed model to a wider audience. Engaging with distinguished international experts and scholars, we strive to continually refine and enhance our models. We eagerly anticipate your continued provision of insightful feedback.

---

## Author Response (AR3)

Dear editors:

Thank you very much for your letter and for the respected reviewers' comments concerning our manuscript entitled "An improved dynamic bidirectional coupled hydrologic-hydrodynamic model for efficient flood inundation prediction" (ID: egusphere-2023-1106). Those comments that the respected editor proposed are all valuable and very helpful for revising and improving our paper, as well as important guiding significance to our research. We have studied comments carefully and have revised the article which we hope meet with approval. There were new lines and page numbers in the revised manuscript. All the changes were marked using red bold in the revised manuscript. We also responded point by point to the reviewers' comments as listed below, along with a clear indication of the revision. Hope these will make it more acceptable for publication.

**Reviewer #1:**

First of all, sincerely thank you very much for your valuable comments. All your suggestions are very important and have important guiding significance for our writing and research. When revising the manuscript, we considered thoughtfully what you have advised.

**1. Comment:** Regarding the mesh generation approach, the authors have opted for a strategy that maintains fine resolutions over the river channel while utilizing coarse resolutions for slope lands. While this approach may have been chosen to optimize computational resources, it raises questions about its suitability for flood modeling, particularly given the traditional emphasis on floodplain inundation processes. The distinction between the proposed d∞ mesh generation method and adaptive mesh generation methods, which typically focus on flow status or topographic differences, warrants clarification. Additionally, the authors should elaborate on their decision to prioritize river flow over floodplain inundation processes, as this choice may impact the comprehensiveness and applicability of the modeling framework.

**Response to comment:** Thank you very much for your valuable comments. Adaptive mesh refinement (AMR) dynamically adapts the grid resolution during the simulation, refining the grid locally based on domain characteristics or flow conditions. AMR is commonly employed in scenarios where flow characteristics exhibit abrupt variations, such as aerodynamic shock waves, hydraulic jumps, and tsunami waves. Capturing discontinuous solutions necessitates local grid refinement, with the location of refinement dynamically adapting to the position of the discontinuities. Consequently, AMR are indispensable. However, AMR needs to segment and merge the grid elements repeatedly during the calculation, which can be time-consuming and offset the calculation time saved by the optimized grid. Besides, the mesh generation and flood simulation were compiled in the same code base, which increased the computation cost and time.

Flow characteristic variations arising from abrupt geometric changes in the computational domain can be captured using static local refinement grids, provided that

the extent of these changes is limited. This approach offers computational time savings. In flood simulations, inundation regions are typically situated in low-lying 2D regions. The outer boundary of the inundation regions can be determined using DEM or calculating by hydrologic models. The D∞ algorithm was employed to preemptively estimate the extent of these areas, providing enhanced computational efficiency relative to AMR and obviating the uncertainty and complexity associated with manual subdivision of the computational domain.

We have introduced the distinction between the proposed D∞ mesh generation method and adaptive mesh generation methods in the revised manuscript (lines 194 to 213). Besides, we have also detailed the applicability and disadvantages of the AMR in introduction of the revised manuscript (lines 98 to 114).

The 1D rivers and 2D inundation regions are prone to flood disasters. In the proposed model, both the 1D rivers and 2D inundation regions were discretized using fine grids, while the coarse grids were used to divide the remain regions. We have elaborated it in the revised manuscript from lines 180 to 186, as follows:

Generally, the river drainage networks present low slopes and hydraulic conveyance, which is subject to flooding. Areas prone to waterlogging, characterized by persistent water saturation, frequently occur adjacent to rivers. The dynamics of inundation in these low-lying zones constitute a central aspect of our investigation. Therefore, these areas should be discretized using fine grids to represent the flooding process in high resolution. However, in the slope lands, fine grids were not required and coarse grids were used to improve computational efficiency.

**2. Comment:** Concerning the efficiency of the improved model (as highlighted in the title), a more extensive demonstration of its efficiency is needed, especially considering the limited area coverage presented in the current results. While the model may indeed offer computational advantages, its effectiveness across larger spatial extents remains unclear. To address this concern, the authors should consider incorporating real-world case studies for model validation, showcasing the model's performance under varying environmental conditions. Furthermore, including the spatial distribution of flood inundation and comparing model outputs with remote sensing-derived flood extent data would enhance the validation process and ensure the model meets the requirements of practical flood modeling applications.

**Response to comment:** Thank you for reading this article carefully and making valuable suggestions. In the Section 3.3 of the revised manuscript, we have simulated the flood processes in a natural watershed, Goodwin watershed. We have verified the numerical accuracy and computational efficiency of the proposed model. The simulated discharge hydrographs were compared with the measured data. In the revised manuscript, we have showed the spatial distribution of flood inundation, as shown in Figure 12. And the computational efficiency was also detailed from lines 594 to 622.

In line with reviewer recommendations, we are endeavoring to apply the model to flood control in real-world. Our research team is currently engaged in flood control projects funded by the Asian Development Bank (ADB), Loan 3485-PRC: Flood control and Environmental Improvement Project in Kongmu River watershed, Xinyu

city, Jiangxi province, China. The proposed M-DBCM is currently being applied to elucidate the effects of reservoir and gate operation, as well as sponge city facilities, on mitigating flood disasters. The M-DBCM has demonstrated superior accuracy in simulating flood inundation in low-lying areas outside the river channel, a capability that is lacking in existing models. Given that the simulation of the Kongmu River watershed pertains to a real-world flood control project, its progress is contingent upon the pace of engineering design. Moreover, the design of flood control infrastructure, including embankments and sponge city facilities, within the Kongmu River watershed has undergone multiple revisions. Consequently, the demands for data collection and processing are comparatively high. Substantial work remains before the model simulation results are ready for publication in academic journals.

During the data collection process for flood simulation, hydrological monitoring stations may experience data loss due to damage or deterioration over time. Efforts are currently underway to address this issue as well. Future work will involve the continued collection of data and information. The model will be employed in diverse scenarios to comprehensively evaluate its performance. In addition, as recommended by the reviewer, a comparison between model outputs and flood extent data obtained from remote sensing could be undertaken.

The proposed model has a relatively short timeframe, typically spanning two to three years. The widespread promotion and application, like Mike series, necessitate a protracted timeframe. Our ultimate objective is to disseminate our research findings through scientific publications, thereby broadening the accessibility and comprehension of the proposed model to a wider audience. Engaging with distinguished international experts and scholars, we strive to continually refine and enhance our models. We eagerly anticipate your continued provision of insightful feedback.

For detailed information regarding the computational efficiency of the model, we have proposed a parameter to quantitatively evaluate the computational efficiency of the M-DBCM (Shen and Jiang, 2023, http://doi.org/10.2166/hydro.2023.131), We defined the evaluation parameter as the ratio of the simulation time of the M-DBCM to that of the full 2D hydrodynamic model (HM2D), as shown in Eq. (1):

$$C = \frac{(1+\alpha_0)t_1 + t_2}{t_0} = \frac{(1+\alpha_0)\alpha \dfrac{A_1}{\Delta x_1^2}\left(\dfrac{T_{end}}{\Delta t_1}\right) + \beta \dfrac{A-A_1}{\Delta x_2^2}\left(\dfrac{T_{end}}{\Delta t_2}\right)}{\alpha \dfrac{A}{\Delta x_1^2}\left(\dfrac{T_{end}}{\Delta t_1}\right)} \tag{1}$$

where $C$ is the assessment parameter to evaluate computational efficiency of M-DBCM; $t_1$, $t_2$ are the computation time on fine and coarse grids, respectively (s); $t_0$ is the

computation time of HM2D (s); $\Delta x_1$, $\Delta x_2$ are the size of fine and coarse grids (m);

$\Delta t_1$, $\Delta t_2$ are the time step on fine and coarse girds (s); $A_1$, $A_2$ are the area of coarse and

fine grids, respectively; $T_{end}$ is simulation time (s); $\alpha$ and $\beta$ are the runtime of hydrodynamic and hydrologic models at one calculation node (s), which is depended on computer power and numerical model complexity. Since the hydrodynamic model is expressed by nonlinear hyperbolic equation and hydrologic model is expressed by linear equation, the calculation of the hydrodynamic model is more complicated than that of the hydrologic model, which results $\dfrac{\beta}{\alpha}<1$.

The time step ratio of coarse grids to fine grids is equal to the size ratio of coarse grids to fine grids, as follows:

$$\frac{\Delta t_2}{\Delta t_1}=\frac{\Delta x_2}{\Delta x_1} \tag{2}$$

Based on Eq. (2), Eq. (1) can be rewritten as:

$$C=\frac{A_1}{A}+\frac{\beta}{\alpha}\left(\frac{\Delta x_1}{\Delta x_2}\right)^3\frac{A-A_1}{A} \tag{3}$$

Define $n=\dfrac{A_2}{A}$ ($0<n\leq1$), $\Delta t_2=k\Delta t_1$ ($k\geq1$), Eq. (3) becomes

$$C=(1-n)+\frac{\beta}{\alpha}\frac{1}{k^3}n \tag{4}$$

From Eq. (4), the computational efficiency of M-DBCM is not only related to the size ratio of coarse to fine grids, but the area ratio of coarse grids to entire domain. If the area of coarse-grid regions are much greater than that of the fine-grid regions, that is, $n\to1$, the assessment parameter becomes $C\propto\dfrac{\beta}{\alpha}\dfrac{1}{k^3}$. It is indicated that the computational efficiency of M-DBCM exponentially improves with the increasing of the area ratio of the coarse grids to entire domain, as shown in Figure 3(a). If the size of coarse grids is much more than that of fine grids, that is, $k\to\infty$, the assessment parameter becomes $C\propto(1-n)$. It is stated that the computational efficiency of M-DBCM improves linearly with the increasing of the size ratio of the coarse to fine grids, as shown in Figure 3(b).

[Figure]

Figure 3 The relationship between the evaluation parameter and the $n$ and $k$:(a) the relationship

between the evaluation parameter and $n$; (b) the relationship between the evaluation parameter and $k$

---

## Author Response (AR4)

Dear editors:

Thank you very much for your letter and for the respected reviewers' comments concerning our manuscript entitled "An improved dynamic bidirectional coupled hydrologic-hydrodynamic model for efficient flood inundation prediction" (ID: egusphere-2023-1106). Those comments that the respected editor proposed are all valuable and very helpful for revising and improving our paper, as well as important guiding significance to our research. We have studied comments carefully and have revised the article which we hope meet with approval. There were new lines and page numbers in the revised manuscript. All the changes were marked using red bold in the revised manuscript. We also responded point by point to the reviewers' comments as listed below, along with a clear indication of the revision. Hope these will make it more acceptable for publication.

**1. Comment:** However, the second comment, which I also emphasized in my editor's decision, is not addressed sufficiently. The comment asks for a validation of your inundation modelling results (spatial extent and depth of flooding) with independent data. You provide evidence that the discharge hydrographs are reproduced well by the model. The key output of your model is inundation maps, and this is precisely the information that should be validated using independent real-world data. This is necessary to demonstrate that your model works for the intended purpose.

**Response to comment:** Thank you very much for your valuable comments. In the Section 3.3 of the revised manuscript, we have simulated the flood processes in a natural watershed, Goodwin watershed. In addition to the discharge hydrographs, we have also showed the spatial distribution of water, as shown in Figure 12 in the revised manuscript. The probability of flooding and inundation increases with increasing water depth. If you want to learn more about this, please review it in the revised manuscript, from lines 580 to 594. However, the inundation maps were not validated due to the lack of real-world flood inundation data. Future work may involve collecting remote sensing data of inundation maps for validation purposes.

The proposed model has a relatively short timeframe, typically spanning two to three years. The widespread promotion and application, like Mike series, necessitate a protracted timeframe. Our ultimate objective is to disseminate our research findings through scientific publications, thereby broadening the accessibility and comprehension of the proposed model to a wider audience. Engaging with distinguished international experts and scholars, we strive to continually refine and enhance our models. We eagerly anticipate your continued provision of insightful feedback.

Our research team is currently engaged in flood control projects funded by the Asian Development Bank (ADB), Loan 3485-PRC: Flood control and Environmental Improvement Project in Kongmu River watershed, Xinyu city, Jiangxi province, China. Several lakes, such as Xianglong Lake and Yudai Lake, are located on one side of the Kongmu River, as shown in Figure 1.

Assuming that the lakes are initially dry, ICM2D simulation results show that some

lakes remain dry under rainstorm frequency with 20-year, as shown in Figure 2. This is inconsistent with the actual situation, as rainfall runoff first flows into the 1D river channels and then into the lakes from the 1D river channels. However, the M-DBCM has demonstrated superior accuracy in simulating flood inundation in low-lying areas outside the river channel (see Figure 3), a capability that is lacking in existing coupling models. Given that the simulation of the Kongmu River watershed pertains to a real-world flood control project, its progress is contingent upon the pace of engineering design. Moreover, the design of flood control infrastructure, including embankments and sponge city facilities, within the Kongmu River watershed has undergone multiple revisions. Consequently, the demands for data collection and processing are comparatively high. Substantial work remains before the model simulation results are ready for publication in academic journals. The flood control project in the Kongmu River watershed is an actual project currently underway in Xinyu city, Jiangxi Province. The results cannot be published without enough discussions and evaluation. Following your suggestions, after further in-depth research, it is our pleasure to submit this application results to the journals.

[Figure]

Figure 1 The location of lakes in the Kongmu river watershed

[Figure]

Figure 2 Preliminary results of flood inundation range obtained from the ICM2D in
Kongmu River Watershed, Xinyu City, Jiangxi Province

[Figure]

Figure 3 Preliminary results of flood inundation range obtained from the M-DBCM